# COMPOSITIONAL ATTENTION:
# DISENTANGLING SEARCH AND RETRIEVAL

**Sarthak Mittal[†], Sharath Chandra Raparthy, Irina Rish, Yoshua Bengio, Guillaume Lajoie[†]**
Mila, Université de Montréal

## ABSTRACT

Multi-head, key-value attention is the backbone of the widely successful Transformer model and its variants. This attention mechanism uses multiple parallel key-value attention blocks (called *heads*), each performing two fundamental computations: (1) *search* – selection of a relevant entity from a set via *query-key* interactions, and (2) *retrieval* – extraction of relevant features from the selected entity via a *value* matrix. Importantly, standard attention heads learn a rigid mapping between search and retrieval. In this work, we first highlight how this static nature of the pairing can potentially: (a) lead to learning of redundant parameters in certain tasks, and (b) hinder generalization. To alleviate this problem, we propose a novel attention mechanism, called *Compositional Attention*, that replaces the standard head structure. The proposed mechanism disentangles search and retrieval and composes them in a dynamic, flexible and context-dependent manner through an additional soft competition stage between the query-key combination and value pairing. Through a series of numerical experiments, we show that it outperforms standard multi-head attention on a variety of tasks, including some out-of-distribution settings. Through our qualitative analysis, we demonstrate that Compositional Attention leads to dynamic specialization based on the type of retrieval needed. Our proposed mechanism generalizes multi-head attention, allows independent scaling of search and retrieval, and can easily be implemented in lieu of standard attention heads in any network architecture.[1]

## 1 INTRODUCTION

Attention mechanisms have become integral parts of machine learning models across a variety of domains. The modern notion of soft-attention was first introduced in Bahdanau et al. (2015) for machine translation to allow recurrent networks to perform well over long sequences. Since then, attention has taken center stage in several models that forego recurrent networks altogether (i.e. Transformers (Vaswani et al., 2017)), and has been leveraged in a wide variety of applications, like natural language (Bahdanau et al., 2015; Vaswani et al., 2017; Devlin et al., 2018), computer vision (Dosovitskiy et al., 2020) and physical reasoning (Ding et al., 2020; Locatello et al., 2020).

At the core of this success is a simple idea: enable task-driven *flexible* connections between elements of a sequence to extract and merge information. This process is implemented by attention (or alignment) functions which, in their simplest form, take a reference or query entity and "pick" (i.e. attend to) the most relevant input entities in a set of other entities. Modern attention systems refine this key principle in two meaningful ways. First, they utilize *key-value attention*, where the attention function takes "queries" from the reference entity, matches them to "keys" attached to input entities, and returns "values" representing a transformation of the selected entities. Second, they allow multiple attention mechanisms to run in parallel, often called *attention heads*, allowing the model to attend to multiple entities jointly, leading to increased expressivity. Despite these advances, Transformer-like architectures still struggle on certain tasks (Fan et al., 2020; Nogueira et al., 2021), including context-sensitive associations and out-of-distribution (OoD) generalization (Lake & Baroni, 2018b; Liska et al., 2018). They are still far from human-level performance on physical reasoning and object-centric tasks (Webb et al., 2020). In an object-oriented world where entities have several attributes, current multi-head attention mechanisms learn rigid search-retrieval associations which lead to various limitations, as illustrated in Figure 1 and Section 2.3.

---

[†]Correspondence authors sarthmit@gmail.com and g.lajoie@umontreal.ca
[1]Open-sourced implementation is available at https://github.com/sarthmit/Compositional-Attention

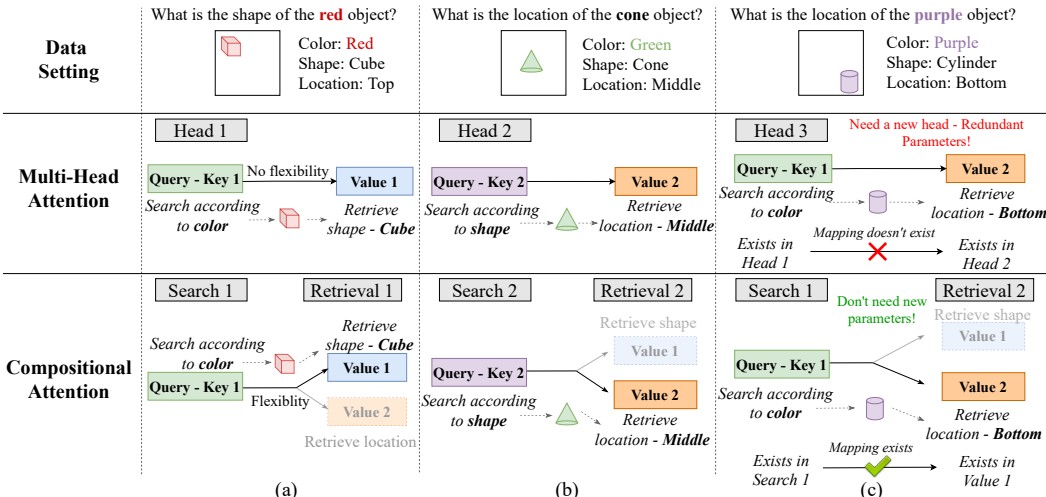

Figure 1: **Motivation behind Compositional Attention.** In a visual question answering setting, we see that the "rigid" mapping between search (query-key) and retrieval (value) in multi-head attention leads to redundant parameters being learned (middle row; (c)). In contrast, when the search and retrieval mechanisms are disentangled and have a pairing set dynamically, these can be composed efficiently without learning of redundant parameters (bottom row; (c)). For details, refer to Section 2.3

Addressing these shortcomings, there are several recent attention-enabled systems developed to allow better decomposition and re-composition of knowledge (Goyal et al., 2019; 2021a;b), some of which we discuss in Appendix A. However, most of these efforts hinge on purpose-built architectural components that remain niche and often are difficult to implement at scale. To complement these efforts and build on the proven efficacy of Transformers, our goal is to develop minimal modifications to key-value attention to enable flexible decomposition of computations found in attention heads, and eliminate some parameter redundancy. Crucially, we aim for a mechanism that is easily implemented and plug-and-play for existing Transformers (and all the models based on them).

We propose *Compositional Attention*, where the search and retrieval operations can be flexibly composed: the key-query search mechanism is no longer bound to a fixed value retrieval matrix, instead it is dynamically selected from a shared pool of value matrices accessible by several *compositional attention heads*. This results in increased flexibility and improved performance.

**Contributions Summary. (a)** We formally describe the shortcomings of rigid search-and-retrieval coupling in standard multi-head attention and empirically analyze them through experiments on an illustrative synthetic task (Section 2.3 and 4.1). **(b)** We present *Compositional Attention* to disentangle search and retrieval, and validate its advantages with a number of experiments (Section 3 and 4 ). **(c)** Through a series of analyses, we demonstrate how our proposed attention mechanism decomposes relational task structure as expected, and facilitates OoD generalization (Section 4). **(d)** We discuss the computational complexity of our proposed method, which can be scaled in either of the components (search and/or retrieval) independently, and is an easy drop-in replacement for multi-head attention in standard Transformer-like architectures (Section 5).

## 2 LIMITATIONS OF MULTI-HEAD ATTENTION

In this section, we first introduce the standard notation for multi-head attention (Vaswani et al., 2017) in terms of search and retrieval mechanisms. We then highlight how the rigidity of the search-retrieval leads to limitations and redundancies in the parametrization of neural networks.

### 2.1 MULTI-HEAD ATTENTION BASICS

**Key-Value Attention:** Given a set of queries and key-value pairs, key-value attention computes a scaled cosine similarity metric between each query and the set of keys. This similarity score determines the contribution of each value in the output for the corresponding query.

More formally, given a set of input elements arranged in a matrix $X \in \mathbb{R}^{N \times d}$, we first obtain queries $Q$, keys $K$ and values $V$ using linear transformations on $X$ with learnable projection matrices

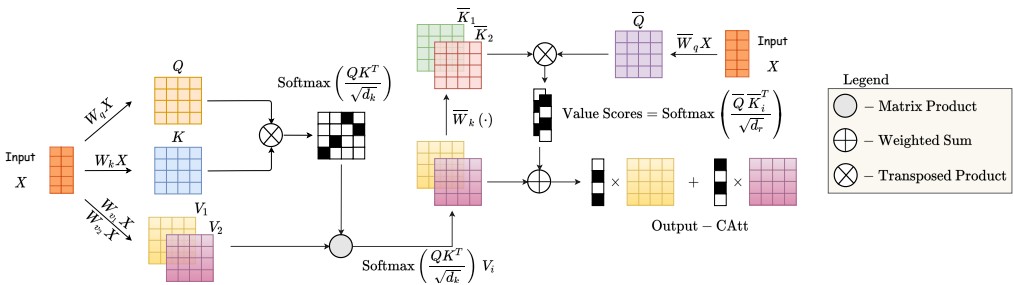

Figure 2: **Computation graph for Compositional Attention.** We show computations for one search and two retrievals. Multiple searches operate in parallel with different search but shared retrieval parameters. The outputs are then fed to a linear network to give the final output as in Equation 14

$W_q \in \mathbb{R}^{d \times d_k}$, $W_k \in \mathbb{R}^{d \times d_k}$ and $W_v \in \mathbb{R}^{d \times d_v}$ respectively. This is given by

$$Q = X W_q \qquad K = X W_k \qquad V = X W_v. \tag{1}$$

For each query, a similarity score is computed with each key using a scaled cosine similarity (called scaled dot-product) to give the attention weights which are used to soft-combine the values as

$$\text{Attention}(Q, K, V) = \text{Softmax}\left(\frac{QK^T}{\sqrt{d_k}}, \text{axis} = \text{'keys'}\right) V \tag{2}$$

where $\frac{1}{\sqrt{d_k}}$ is the scaling factor.

**Multi-Head Attention:** A multi-head attention mechanism combines multiple (say, $h$) independent key-value attention mechanisms in parallel to provide the model the ability to jointly attend to different positions and hence increase representational capacity. The outputs resulting from these multiple heads are concatenated together and then linearly projected back to the input dimension using a learnable matrix $W^o \in \mathbb{R}^{h d_v \times d}$:

$$\text{Multi-Head} = \text{Concat}\Big(\text{head}_1, \text{head}_2 \dots \text{head}_h\Big) W^o \tag{3}$$

where $\text{head}_i = \text{Attention}(Q_i, K_i, V_i)$.

## 2.2   Search and Retrieval Components

Here, we take the multi-head attention defined in Section 2.1 and decompose it into its two fundamental components - *Search* and *Retrieval*.

**Search:** A search is parameterized by the query and key matrices, that is, $W_q$ and $W_k$ respectively. These parameters define a notion of compatibility metric between pairs of element $x_j$ and $x_k \in X$:

$$\text{Search}\Big(Q, K\Big) = \text{Softmax}\left(\frac{QK^T}{\sqrt{d_k}}, \text{axis} = \text{'keys'}\right) \tag{4}$$

where $Q = X W_q$ and $K = X W_k$. The above computation gives the compatibility between an element $x_j$ with other elements $x_k$'s under the compatibility metric defined by the *Search* parameters.

**Retrieval:** A retrieval is parameterized by a value matrix $W_v$ describing the kind of features in the input elements in $X$ that are relevant and need to be accessed for the downstream task:

$$\text{Retrieval}\Big(\text{Search}\big(Q, K\big), V\Big) = \text{Search}\Big(Q, K\Big) V \tag{5}$$

where $V = X W_v$. Note that each Retrieval defines the kind of attributes to access from input $x_k'$s and can take any Search result as its input.

**Multi-head Attention as a rigid pairing of Searches and Retrievals:** Given the above definitions, one can see how standard multi-head attention amounts to a rigid pairing of Searches and Retrievals, such that an end-to-end function of fixed attribute pairs are learned at optimization. Indeed, $h$ heads are composed of $h$ different searche-retrieval pairs – the $i^{th}$ retrieval is performed only on the $i^{th}$ search. Multi-head attention thus amounts to a special case of Equation 4 and 5

$$\text{head}_i = \text{Retrieval}\Big(\text{Search}(Q_i, K_i), V_i\Big) \qquad (\textit{Note: Same subscript } i) \tag{6}$$

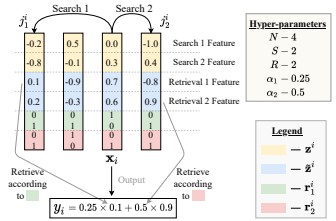 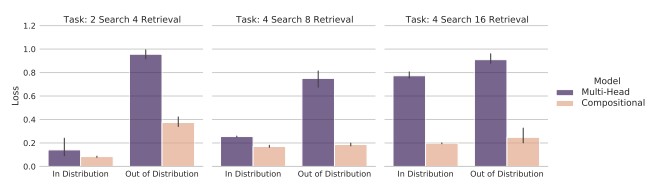

Figure 3: **Left: Contextual Retrieval Task Illustration.** Dynamic search and retrieval based on search, retrieval and retrieval context features. Each element has a corresponding output but we show it only for $\mathbf{x}_i$ for brevity. **Right: Performance on Contextual Retrieval Task.** Here, we compare our proposed model against standard Multi-Head attention model (lower is better) on various setups of the task. Our proposed model outperforms the baseline in both in-distribution as well as out-of-distribution settings.

Viewing multi-head attention through these fixed search-retrieval pairings foreshadows a possible generalization of searches and retrievals which we propose below. Before doing so, however, we first highlight specific shortcomings of standard multi-head attention.

## 2.3 Shortcomings of rigid associations

As described in Section 2.2, multi-head attention considers a fixed pairing between searches and retrievals. While it has been widely successful across a variety of domains, we hypothesize that this rigid mapping is not always ideal and can sometimes lead to reduced capacity and learning of redundant parameters, missing an opportunity for better systematic generalization. We note that the search associated with each head defines a feature (defined by query-key matrices $W_q$ and $W_k$) based on which compatibility between objects is evaluated. Further, each head's retrieval allows the model to access a particular feature (defined by value matrix $W_v$) from the searched objects. Following this, we showcase two types of redundancies that can arise in multi-head attention: (a) *Search Redundancy* which leads to learning of redundant query-key matrices and (b) *Retrieval Redundancy* which leads to learning of redundant value matrices.

We highlight these two redundancies jointly using a simple example illustrated in Figure 1, where three objects with attributes: shape, color and location, are the subject of different questions. In **(a)** the model has to learn to search according to *color* and correspondingly retrieve *shape* information while in **(b)** it has to search according to *shape* and retrieve *location*. On this task, standard multi-head attention (middle row) should learn two heads, one each for **(a)** and **(b)**. To answer the question in **(c)**, the model has to search according to *color* and retrieve *location*. While searching according to color exists in **(a)** learned by *Head 1* and retrieving location exists in **(b)** learned by *Head 2*, there is no way to combine them. Thus, another head is needed to obtain the search of *Head 1* and retrieval of *Head 2*. This leads to parameter redundancy and a missed opportunity to factorize knowledge more efficiently, since these pieces of learned knowledge individually exist in *Head 1* and *Head 2* already.

The scenario in Figure 1 may look highly idealized because multi-head attention might not limit searches/retrievals on single features and is capable of doing more nuanced soft-combinations. While this may be the case for this simple example, what it highlights is the danger of rigid learned associations that limits re-composition of learned pieces of knowledge, leads to redundant parameters and potentially limits OoD generalization, irrespective of what the model learns. We discuss this in more detail in Appendix B.1, and empirically explore these principles in a purpose built diagnosis task we call *Contextual Retrieval Task*, in Section 4.1. Finally, we reiterate that multi-head attention with $h$ heads can only represent up to $h$ unique (Search – Retrieval) pairings. In what follows, we propose to alleviate this fundamental limitation by allowing for $S \times R$ such pairings, with $S$ the number of search types and $R$ the number of retrieval types.

## 3 Compositional Attention - Disentangling Search and Retrieval

We propose a novel attention mechanism that relaxes static search-retrieval pairing in favour of a more flexible and dynamic mapping. To do this, we let go of the concept of "head" altogether and instead propose independent and recombinable *Searches* and *Retrievals*, as defined in Section 2.2. The central innovation lies in the way these two components are combined: with query-key attention on retrievals.

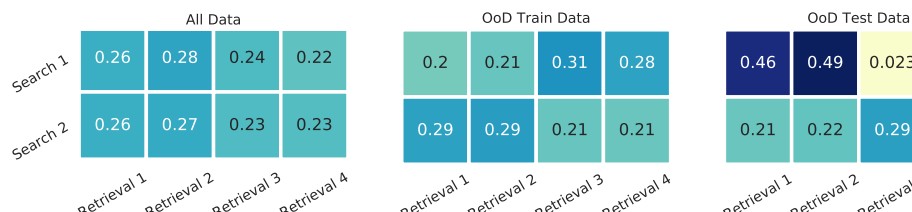

Figure 4: **Efficient Composition in Contextual Retrieval Task.** We plot the average search-retrieval activation statistics across data with **Left:** All possible value combinations, **Middle:** Subset of value combinations used for training, and **Right:** Remaining subset of value combinations used for OoD testing. The activation statistics switch distinctly between OoD training and testing and stay around the average when all possible subsets are shown, thus highlighting good specialization.

Similar to heads, we start by defining $S$ parallel search mechanisms. That is, we have $S$ different query-key parameterizations $W_{q_i}$ and $W_{k_i}$ respectively. The output of each search is as defined in Equation 4. In essence, for each search $i$, we get

$$Q_i = X\,W_{q_i} \qquad \text{and} \qquad K_i = X\,W_{k_i} \tag{7}$$

$$\text{Search}_i = \text{Softmax}\left(\frac{Q_i K_i^T}{\sqrt{d_k}}\,,\,\text{axis} = \text{`keys'}\right) \tag{8}$$

Next, we define $R$ different retrieval mechanisms, which correspond to the $R$ different $W_{v_j}$ matrices. These matrices are used to obtain different attributes from the input. Formally, this is summarized as

$$V_j = X\,W_{v_j} \tag{9}$$

where $V_j$ highlights accessing of different attributes. Then, corresponding to each search, all possible retrievals are done. This is similar to Equation 5 and is defined as

$$\text{Retrieval}_{ij} = \text{Search}_i\,V_j \tag{10}$$

for all $i, j$. This step gives us all the hypothetical retrievals for each search and out of this, one retrieval per search needs to be instantiated. This instantiation is done through a secondary attention mechanism computed using retrieval queries $\overline{Q}_i$ and retrieval keys $\overline{K}_{ij}$, that are obtained as

$$\overline{Q}_i = X\,\overline{W}_{q_i} \qquad\qquad \overline{K}_{ij} = \text{Retrieval}_{ij}\,\overline{W}_k \tag{11}$$

where the parameter $\overline{W}_{q_i} \in \mathbb{R}^{d \times d_r}$ is a different matrix for each search indexed by $i$ and, together with $\overline{W}_k$, is tasked with driving the pairing between search and retrieval. We broadcast the matrix $\overline{Q}_i$ from $\mathbb{R}^{N \times d_r}$ to $\mathbb{R}^{N \times 1 \times d_r}$ and define $\overline{K}_i \in \mathbb{R}^{N \times R \times d_r}$ by

$$\overline{K}_i = \text{Concat}\left(\overline{K}_{i1}, \overline{K}_{i2}, \ldots, \overline{K}_{iR}\right). \tag{12}$$

Thus, through these retrieval queries and keys, the required instantiation per search is done as

$$\text{CAtt}_i = \underbrace{\text{Softmax}\left(\frac{\overline{Q}_i \overline{K}_i^T}{\sqrt{d_r}}\,,\,\text{axis} = \text{`retrieval'}\right)}_{\text{Value Scores}} \text{Retrieval}_{ij} \tag{13}$$

where the transpose is over the last two axes. Hence, for each search $i$, the softmax gives attention weights over all possible retrievals and the winning retrieval is instantiated through this soft attention mechanism. Finally, similar to multi-head attention, the outputs of these parallel searches are combined by concatenating them and passing them through a linear network:

$$\text{CompositionalAttention} = \text{Concat}\left(\text{CAtt}_1, \text{CAtt}_2, \ldots, \text{CAtt}_S\right)W^o \tag{14}$$

where $W^o \in \mathbb{R}^{Sd_v \times d}$. Note that in this mechanism, the choice of retrieval for each search is not fixed, and instead is dynamically modulated by $\overline{Q}_i$ and $\overline{K}_i$ respectively. We refer the readers to Figure 2 for a visual depiction of the computation graph.

Compositional Attention allows the model to have **(a)** Different number of searches and retrievals, i.e. $S$ and $R$ respectively, **(b)** Dynamic selection of shared retrievals for each search and **(c)** Representation capacity of $S \times R$ (Search – Retrieval) pairings. Thus, we highlight that *Compositional Attention* disentangles search and retrieval and solves the redundancies of multi-head attention (Section 2.3).

| Algorithm | Searches | Retrievals | Unary Accuracy | Binary Accuracy | Ternary Accuracy |
|---|---|---|---|---|---|
| Transformer | 4 | 4 | $98.6_{\pm 0.2}$ | $84.4_{\pm 5.3}$ | $64.9_{\pm 3.3}$ |
| | 8 | 8 | $98.5_{\pm 0.2}$ | $84.5_{\pm 6.0}$ | $65.4_{\pm 4.7}$ |
| Compositional Transformer | 4 | 1 | $98.7_{\pm 0.2}$ | $86.8_{\pm 2.8}$ | $66.4_{\pm 1.3}$ |
| | | 2 | $98.8_{\pm 0.1}$ | $88.2_{\pm 3.2}$ | $66.9_{\pm 1.8}$ |
| | | 3 | $98.9_{\pm 0.2}$ | $89.8_{\pm 1.1}$ | $67.1_{\pm 1.5}$ |
| | | 4 | $98.6_{\pm 0.3}$ | $84.9_{\pm 4.5}$ | $64.8_{\pm 4.1}$ |

Table 1: **Performance on Sort of CLEVR.** We highlight that our proposed model outperforms the baseline across the different question types even with lower number of searches and/or retrievals.

## 4 EXPERIMENTS

For all our experiments, we consider the standard Transformer model (Vaswani et al., 2017) with *parameter sharing* across layers (Dehghani et al., 2018) as our baseline. For visual tasks, we consider Vision Transformer introduced by Dosovitskiy et al. (2020) as our baseline. Our proposed model, *Compositional Transformer*, uses the *Compositional Attention* mechanism as a drop-in replacement for the multi-head attention block while keeping the rest of the architecture same. We also perform ablations on number of retrievals, model sizes and complexities as discussed in Appendix B.7

Through our experiments, we show that **(a)** flexible composition of search and retrieval leads to better in distribution and out of distribution performance, and **(b)** our proposed model can achieve similar or better performance, often with fewer retrievals.

### 4.1 CONTEXTUAL RETRIEVAL TASK

As a start, we consider a purpose built task to better understand how learned attribute mappings may be recombined by attention mechanisms. Our goal in introducing this experiment is to explicitly control search and retrieval attributes that allows for **(1)** systematic testing for OoD generalization, and **(2)** direct comparisons between expected/ground-truth and learned activation patterns. Thus, we propose a supervised set-based regression task where objects with several features need to be selectively associated with one another (e.g. find the closest blue object), and distinct feature values must be returned based on inherent contexts (e.g. report shape if position is middle and size otherwise). Our task seeks a minimal setting where such associations can be present, and involves a collection of $N$ objects $\{\mathbf{x}_i\}_{1=1}^{N}$, each with scalar-valued features, as illustrated in Figure 3 (*left*). Corresponding to each object $\mathbf{x}_i$, the output is a scalar

$$y_i = \sum_{s=1}^{S} \alpha_s v_s^i, \tag{15}$$

where $\alpha_s$ are randomly selected fixed weights, and $v_s^i$ are the results of $S$ comparative searches pairing object $\mathbf{x}_i$ with other objects in the set. To describe these comparative searches, we first outline the composition of each object. Here,

$$\mathbf{x}_i = \{(z_1^i, \ldots, z_S^i), (\tilde{z}_1^i, \ldots, \tilde{z}_R^i), (\mathbf{r}_1^i, \ldots, \mathbf{r}_S^i)\}, \tag{16}$$

where $\mathbf{z}^i \in \mathbb{R}^S$ and $\tilde{\mathbf{z}}^i \in \mathbb{R}^R$ contain $S$ "search" features and $R$ "retrieval" features respectively, and one-hot vectors $\mathbf{r}_s^i \in \{0,1\}^R$ contain the *retrieval preference* of object $\mathbf{x}_i$ for search $s$.

*Search Step:* Each object $\mathbf{x}_i$ searches for $S$ objects in parallel that are closest to it based on the search feature. This operation is given by

$$j_s^i = \arg\min_{j \neq i} \left| z_s^i - z_s^j \right| \tag{17}$$

where $s$ denotes the $s^{th}$ search feature, and hence there are $S$ parallel independent searches with $j_s^i$ denoting the winner of search $s$ for object $\mathbf{x}_i$.

*Retrieval Step:* Corresponding to each winner $j_s^i$, the retrieval context $\mathbf{r}_s^i$ dictates which retrieval feature to access from the $j_s^i$ object. This is given by

$$v_s^i = \tilde{z}_{\mathbf{r}_s^i}^{j_s^i} \tag{18}$$

| Algorithm | Searches | Retrievals | CIFAR10 | FashionMNIST | SVHN | Equilateral Triangle |
|---|---|---|---|---|---|---|
| Transformer | 4 | 4 | $77.2_{\pm0.3}$ | $89.4_{\pm0.0}$ | $85.0_{\pm0.1}$ | $96.8_{\pm0.1}$ |
| Compositional Transformer | 4 | 1 | $77.5_{\pm0.5}$ | $89.9_{\pm0.0}$ | $85.0_{\pm0.3}$ | $97.0_{\pm0.0}$ |
| | | 2 | $77.9_{\pm0.2}$ | $89.9_{\pm0.4}$ | $86.0_{\pm0.7}$ | $97.2_{\pm0.3}$ |
| | | 3 | $78.0_{\pm0.1}$ | $89.9_{\pm0.4}$ | $85.2_{\pm0.3}$ | $97.1_{\pm0.4}$ |
| | | 4 | $77.6_{\pm0.2}$ | $90.0_{\pm0.0}$ | $86.0_{\pm0.5}$ | $96.9_{\pm0.6}$ |

Table 2: **Performance on Multi-Task Image Classification.** We highlight that our proposed model outperforms the baseline across different number of retrievals.

Training is done by minimizing the $L_1$ loss on $\{y_i\}_{i=1}^N$ targets (see Appendix C.1 for details). While seemingly synthetic, this task considers multiple objects with various attributes and tests the model's ability to flexibly compose searches and retrievals. This is tightly associated with real world scenarios but in a controlled low dimensional setting. We provide a visualization of the task in Figure 3 (*left*).

**OoD Setup:** Since all searches share the different values to be retrieved, we construct an out-of-distribution version of the dataset by showing only certain ground truth search-retrieval combinations in training and others during testing. More specifically, for a task with $S$ searches and $R$ retrievals there are $R^S$ possible unique value combinations which define the *retrieval preference*. For the OoD setup, we use a fraction of them in the training set and the rest in the test set. We stress that only models that factorize search and retrieval efficiently and compose them flexibly can do well on the unseen search-retrieval combinations.

**Quantitative Results:** We experiment on a range of searches $S$ and retrievals $R$. Since this task only requires one set of parallel searches, we use one-layered attention models for experiments. We observe that compositional attention consistently outperforms multi-head attention across a wide range of task hyperparameters in both in-distribution as well as OoD settings (Figure 3 – *right*). Further, our proposed method outperforms the baseline across various model complexities.

**Analysis:** We visualize the value scores from Equation 13, aggregated over all searches and entities and contrast them with ground-truth retrievals that the task provides. Figure 5 shows that the model specializes on what features to retrieve. We also analyse the search-retrieval pairings by taking a model trained on a subset of all possible value combinations and plotting its activation statistics across **(1)** all possible value combinations, **(2)** value combinations seen during training, and **(3)** value combinations seen during OoD testing in Figure 4. As expected, we see that activation statistics average out in **(1)**, as each search features are called on in a uniform i.i.d. manner. In contrast, we see increasing specialization of activation pattern for **(2)** and **(3)**, respectively, consistent with selective recombination made possible by compositional attention. The key finding from this analysis is that compositional attention is dynamically specializing over value retrievals, allowing for better generalization on unseen value combinations in the OoD setup.

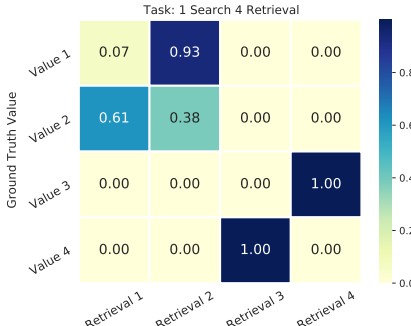

Figure 5: **Retrieval Specialization on Contextual Retrieval Task.** The proposed model learns to specialize its own retrieval (X-axis) based on ground truth values (Y-axis).

Appendix C.1 contains further details about the task, hyperparameters and further ablations on model complexities and specialization visualizations across different task settings.

## 4.2 Relational Reasoning in Sort-of-CLEVR

Sort-of-CLEVR (Santoro et al., 2017) is a Visual Question-Answering (VQA) task that tests the model's understanding of the scene by posing questions based on the properties of the objects and their relations with each other. The task consists of three types of questions: (a) *Unary*, which are based on properties of single objects, (b) *Binary*, which are based on the relationship between two objects, and (c) *Ternary*, which are based on relationship between three objects.

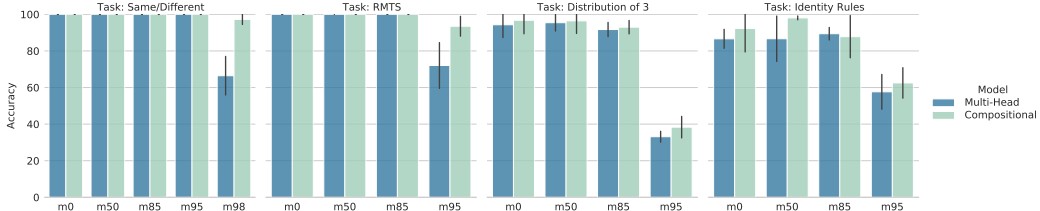

Figure 6: **Performance on ESBN Tasks.** Our proposed model outperforms the baseline across different tasks especially in the extreme OoD setting.

**Quantitative Results:** We use 4-layered transformer models with parameter sharing for our experiments. Table 1 highlights that Compositional Transformer outperforms standard transformers across different question types (unary, binary and ternary), even with fewer searches and/or retrievals than the number of heads in the standard transformer. This is attributed to the fact that our model is exploiting the inherent compositional nature of this task by reusing the parameters for dynamic retrieval of context dependent attributes.

We refer the readers to Appendix C.2 for details regarding this task, model hyperparameters and performance ablations on a variety of model settings.

### 4.3 EQUILATERAL TRIANGLE DETECTION

This is a binary classification task introduced by Ahmad & Omohundro (2009), where the aim is to decide if the set of three point clusters in the image forms an equilateral triangle. For a triangle to be equilateral, the midpoints of these clusters should be equidistant from each other.

| Algorithm | Searches | Retrievals | Test Accuracy |
|---|---|---|---|
| Transformer | 4 | 4 | $93.8_{\pm 0.1}$ |
| Compositional Transformer | 4 | 1 | $95.6_{\pm 1.2}$ |
| | | 2 | $96.7_{\pm 0.4}$ |
| | | 4 | $97.0_{\pm 0.3}$ |

**Quantitative Results.** Table 3 highlights that compositional attention outperforms multi-head attention across different number of retrievals.

Table 3: **Performance on Equilateral Triangle Detection.** Our proposed method outperforms the baseline over different number of retrievals.

We refer the readers to Appendix C.3 for more details.

### 4.4 MULTI-TASK IMAGE CLASSIFICATION

We pose the problem of image classification across four different datasets – CIFAR10, FashionMNIST, SVHN and Equilateral Triangle Detection as a multi-task learning setup. To perform well on this, models need to figure out which information is shared across datasets and which is private to a dataset.

**Quantitative Results:** We train a 2-layered universal transformer with four learnable embeddings, one for each task. Our proposed model replaces the multi-head attention with compositional attention in the baseline. Table 2 shows that the proposed model outperforms the baseline.

Appendix C.4 contains further details about the task and the models.

### 4.5 LOGICAL REASONING IN ESBN TASKS

Webb et al. (2020) introduces a suite of four tasks to test the model's ability to perform logical reasoning in an OoD setting. Within each task, the input is a sequence of objects and the output is based on the relationships between these objects. For example, in the "Same/Different" task the model receives two objects and has to predict whether the objects are same or different. Each task has OoD criteria – eg. m95 refers to training with 5 out of 100 objects and testing with the remaining 95.

**Quantitative Results:** We use a single layered transformer model as was done in the original benchmark (Webb et al., 2020). Figure 6 highlights that Compositional Attention outperforms Multi-Head Attention across all the tasks, especially at higher hold-out (m) values. This shows that the proposed model is able to better handle distribution shifts than the baseline model across all tasks.

We refer the reader to Appendix C.5 for details regarding the tasks as well as the models.

| Algorithm | Searches | Retrievals | Cutoff Length | | | | | | | |
| --- | --- | --- | --- | --- | --- | --- | --- | --- | --- | --- |
| | | | 22 | 24 | 25 | 26 | 27 | 28 | 29 | 30 |
| Transformer | 8 | 8 | $0.01_{\pm0.01}$ | $0.06_{\pm0.02}$ | $0.12_{\pm0.04}$ | $0.11_{\pm0.06}$ | $0.22_{\pm0.08}$ | $0.02_{\pm0.04}$ | $0.03_{\pm0.06}$ | $0.05_{\pm0.08}$ |
| Compositional Transformer | 8 | 1 | $0.01_{\pm0.01}$ | $0.03_{\pm0.02}$ | $0.24_{\pm0.07}$ | $0.34_{\pm0.08}$ | $0.38_{\pm0.18}$ | $0.10_{\pm0.11}$ | $0.13_{\pm0.10}$ | $0.08_{\pm0.06}$ |
| | | 2 | $0.01_{\pm0.02}$ | $0.04_{\pm0.04}$ | $0.24_{\pm0.16}$ | $0.27_{\pm0.13}$ | $0.49_{\pm0.09}$ | $0.09_{\pm0.09}$ | $0.09_{\pm0.08}$ | $0.16_{\pm0.12}$ |
| | | 4 | $0.00_{\pm0.00}$ | $0.05_{\pm0.02}$ | $0.06_{\pm0.03}$ | $0.30_{\pm0.14}$ | $0.26_{\pm0.13}$ | $0.04_{\pm0.04}$ | $0.11_{\pm0.09}$ | $0.08_{\pm0.11}$ |
| | | 8 | $0.02_{\pm0.02}$ | $0.05_{\pm0.02}$ | $0.14_{\pm0.05}$ | $0.18_{\pm0.10}$ | $0.30_{\pm0.08}$ | $0.04_{\pm0.05}$ | $0.06_{\pm0.11}$ | $0.08_{\pm0.08}$ |

Table 4: **Performance on SCAN.** We highlight that our proposed model outperforms the baseline across the different question types even with lower number of searches and/or retrievals.

## 4.6 SCAN DATASET

SCAN (Lake & Baroni, 2018a) is a synthetic sequence to sequence task aimed at translating natural language instructions into a sequence of actions for a robot to follow. We follow the length extrapolation generalization split (Newman et al., 2020; Csordás et al., 2021) where training is done on lengths of 22 output actions and tested on larger output sequence lengths.

**Quantitative Results:** We train a 3-layered universal transformer as the baseline and compare it with our proposed plug-in attention replacement. Table 4 showcases that compositional attention outperforms the standard multi-head attention across multiple cutoff lengths.

We refer the readers to Appendix C.6 for further details regarding the task and models.

## 4.7 LANGUAGE MODELLING

We perform experiments on the WikiText-103 data corpus (Merity et al., 2016) for the language modeling task. Here, the task is to predict probabilities for next or masked words, evaluated through perplexity.

**Quantitative Results:** We use 6-layered transformer models with parameter. We plot the *validation perplexity* against epochs in Figure 7 which highlights that our proposed attention mechanism not only outperforms the baseline but also converges faster. Further, we see that our proposed model obtains *test perplexity* $38.8_{\pm0.0}$ as opposed to baseline's perplexity $39.6_{\pm0.3}$.

We refer the reader to Appendix C.7 for further details.

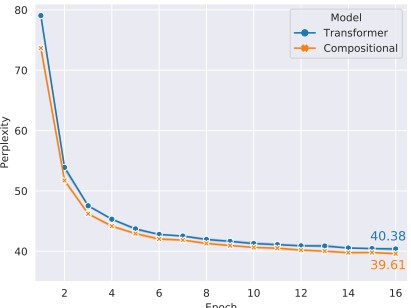

Figure 7: **Performance on Language Modeling (WikiText103).** We illustrate that our proposed mechanism outperforms the standard multi-head attention.

## 5 DISCUSSION AND CONCLUSION

**Summary.** In this work, we revisit Multi-Head Attention, a popular attention mechanism, and highlight its shortcomings due to the rigid association between search and retrieval mechanisms. We argue that this rigid coupling hinders re-usability of parameters and reduces the expressivity of the model. To mitigate this, we propose a novel mechanism which uses a value retrieval mechanism to flexibly compose searches and retrievals. Experiments on various tasks show that our proposed method outperforms standard multi-head transformers, while often using only a fraction of retrievals.

**Complexity.** While our proposed mechanism requires additional parameters for the computation of value scores, we highlight that this increase is often minuscule compared to the total number of parameters. Crucially, we note that this light increase in parameters per search mechanism is easily offset by reducing the number of retrievals needed. For all our experiments, our proposed models offer similar capacity as the baselines unless stated otherwise. This highlights that the improved performance is due to flexible composition of search and retrieval and not number of parameters. We discuss computational complexity in detail in Appendix B.5.

**Limitations and Conclusion.** Motivated by the need for efficient factorization of knowledge and dynamic reusability of learned pieces of computations, we propose *Compositional Attention*, a first step towards flexible composition of search and retrieval. We also highlight some of the limitations of the proposed methodology in Appendix B.8 and hope that our work would promote research into more flexible models capable of better systematic generalization.

## ACKNOWLEDGEMENTS

The authors would like to thank Felipe Codevilla, Nicolas Gontier, Disha Shrivastava, Damjan Kalajdzievski, Olexa Bilaniuk, Ioannis Mitliagkas and Gauthier Gidel for their time and effort on helping improve this work. SM would like to acknowledge the support of UNIQUE and IVADO towards his research. IR, YB and GL acknowledge the support from Canada CIFAR AI Chair Program, as well as Samsung Electronics Co., Ldt. IR acknowledges support from the Canada Excellence Research Chairs Program. GL acknowledges NSERC Discovery Grant [RGPIN-2018-04821].

## ETHICS STATEMENT

The authors do not foresee negative societal or ethical impacts beyond those expected from general improvements in ML. Furthermore, the authors note that with added flexibility comes the increased possibility of learning negative implicit biases in real-world applications. Much like other cutting edge ML methods, careful considerations of potential negative impacts on society should be considered before deploying models for applied uses.

## REPRODUCIBILITY STATEMENT

The authors provide the code used to run all the required experiments and will open source their code with ample documentation to allow for ease of reproducibility. Further, the authors provide the exact formulation of the proposed model in Section 3 and provide the hyperparameters as well as different existing codebases used for all the experiments in Appendix C.

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

# APPENDIX

## A    RELATED WORK

The advent of transformer-like models have led to advancements on various flavours of attention based models. This revolution first started with augmenting Recurrent Neural Networks (RNNs) with a form of semi-parametric memory structure through attention (Bahdanau et al., 2015) and it soon led to people questioning the need for recurrence. This line of questioning resulted in a famous class of models that get rid of recurrence in favour of just parallel self-attention computations that are quite efficient to do on modern hardware (Vaswani et al., 2017). We briefly discuss the various advances along these lines and distinguish how our proposed attention algorithm is different from them.

### A.1    ATTENTION

Attention has been a major component of human cognition which allows humans to selectively process relevant information from the plethora of sensory stimulus we receive. The idea of selecting relevant features from a sea of information allows us to make predictions in both a robust as well as compute efficient way. Inspired from neural cognition, there have been a lot of efforts in trying to introduce a notion of attention to relevant states of the input for reliable downstream prediction (Xu et al., 2015; Luong et al., 2015; Kerg et al., 2020).

A major problem in Recurrent Neural Networks based systems is the problem of vanishing and exploding gradients that happens due to improper credit assignment in the model. This is because RNNs model all the information seen up to a certain time through a parametric fixed sized vector which undergoes repeated computations over all time steps. This makes the system brittle to changes in sequence lengths or in presence of long sequence of distracting information. A way to solve this problem was to move away from parametric representations of the entire past and instead rely on dynamic semi-parametric "memory" to allow these models to look back whenever needed (Graves et al., 2014; Bahdanau et al., 2015). These works aimed at augmenting recurrence with self-attention and demonstrated that when combined with these cognition-inspired inductive biases, ML systems were able to extrapolate much better to larger sequence lengths.

Following this, there has been a lot of recent work that then aimed to remove recurrence between time-steps and rely solely on querying information through self-attention. Recent advances on multiple domains (Vaswani et al., 2017; Dosovitskiy et al., 2020; Ding et al., 2020; Locatello et al., 2020) showcased that removing recurrence from the picture and relying solely on parallel computations not only leads to significant improvements in performance and generalization but is also easier and faster to train on current hardware. Since the advent of these transformer based models built fundamentally on multi-head attention, the role of attention has become increasingly important across various domains like vision, language and reinforcement learning. It has also led to a lot of research on various architectural choices in fully attention-based systems, some of which we discuss in Appendix A.2.

It is, however, important to note that there has been some research that highlight the need for recurrence jointly with self-attention for solving certain logical reasoning tasks efficiently (Hudson & Manning, 2018; Selvakumar et al., 2018; Webb et al., 2020).

### A.2    TRANSFORMER VARIANTS

The ubiquity of self-attention models in the current ML community has led to tremendous research aimed at incorporating different inductive biases in the attention mechanism used; namely in the multi-head attention. Most of these variants aim to alter multi-head attention in a way that would remove the quadratic time complexity computational bottleneck that is present in standard multi-head attention. However, there are certain works that aim more on the fundamental inductive biases that the attention encodes as opposed to computational benefits. We discuss some of these variants here.

**Reducing Computational Complexity.** Given a set of $n$ vectors, the standard multi-head attention aims to create an $n \times n$ attention matrix that takes quadratic complexity to compute. This bottleneck prevents usage of self-attention when $n$ is large. In light of this, a lot of recent research aims to reduce this quadratic complexity to $n \log n$ or linear complexity. This is often achieved by either introducing some restrictions in the $n \times n$ attention matrix through locality sensitive hashing (Kitaev et al., 2020),

sparsity (Child et al., 2019), low rank approximation (Wang et al., 2020) or through random features for approximation of softmax (Choromanski et al., 2020). We refer the readers to Tay et al. (2020) for a more detailed analysis of different transformer variants.

**Additional Inductive Biases.** While a lot of the above transformer variations are designed to prevent the quadratic bottleneck, most of them also add certain additional inductive biases in the model. For example, the addition of sparsity not only reduces the computational complexity but also adds the additional inductive bias of sparse information routing between different elements. There are certain additional variants (Lamb et al., 2021; Goyal et al., 2021b) that add other inductive biases, eg. factorized state space and global workspace bottleneck respectively in the transformer model.

### A.3 MODULARITY, COMPOSITIONALITY, REUSABILITY AND BOTTLENECK

There have been recent efforts along the lines of modularized computations in an effort to improve the model's capacity to perform systematic generalization. In particular, humans are able to compartmentalize information and act on it in a disentangled, context-driven and robust fashion. These cognitive fundamentals have led to a preliminary movement of Machine Learning systems into this space. We discuss some of the essential ingredients below.

**Modularity.** Modularity refers to factorization of knowledge into smaller components that can independently exist and act on sensory information. It can be considered as disentangled representations that allow for interventions on these different components or factorized mechanisms where each mechanism has a specific purpose and can act on a part or whole of the sensory information. The fundamental aim of modularity is to prevent unrestricted information flow across a whole monolitihic system and instead to learn in an often end-to-end fashion factorized representations and mechanisms that act on these representations. Recent works (Goyal et al., 2019; 2020; 2021a; Mittal et al., 2020; Madan et al., 2021; Lamb et al., 2021; Ke et al., 2021) along the lines of factorizing knowledge demonstrate that it often leads to increased robustness and better OoD performance.

**Compositionality and Reusability.** Humans are able to perform complex tasks even in novel and unknown situations. This capacity often stems from the fact that our complex actions are in reality compositions of simpler primitives and our understanding of these primitives is so good that we are able to dynamically combine these primitives into novel complex actions. Recent research has started looking into tasks and systems that test and allow for compositional generalization (Lake & Baroni, 2018a; Li et al., 2019; Keysers et al., 2019; Chen et al., 2020; Hupkes et al., 2020; Goyal & Bengio, 2020), which is generalization to novel combinations of the underlying primitives/mechanisms. The primary reason why a number of modular systems are constructed in recurrent domains is because we want the factorized mechanisms to be reusable in a number of scenarios. Reusability of knowledge (Dehghani et al., 2018; Bai et al., 2019) allows for learning of disentangled mechanisms in a modular system which then has the potential to lead to efficient compositions of the learned disentangled mechanisms. Recent success of systems that use computations that can be reused multiple times demonstrates that reusability is actually an important fundamental for obtaining compositionality.

**Bottleneck.** Conscious attention in humans is a key ingredient to create a bottleneck of information processing, according to the Global Workspace Theory (Baars, 1997; Dehaene et al., 2017). The key use of this bottleneck is to restrict information flow across the whole network, human brain or otherwise, which allows for robustness to insignificant pieces of sensory information. The usefulness of this bottleneck has been hypothesized to be linked to the sparsity and simplicity of the dependencies manipulated with System 2 cognition (Bengio, 2017; Goyal & Bengio, 2020). Recent works along these lines (Goyal et al., 2021b) illustrate that modular systems with the addition of a bottleneck efficiently factorize computations and then compose them in a dynamic and context dependent fashion often lead to improved performance, faster adaptation and systematic generalization (Bengio et al., 2019; Ke et al., 2021).

## B PROPOSED MODEL

In this section, we provide additional details about the general motivation, architecture setup and our argument for using parameter sharing across layers. We further provide details about computational complexity of the proposed model and some ablations that we consider.

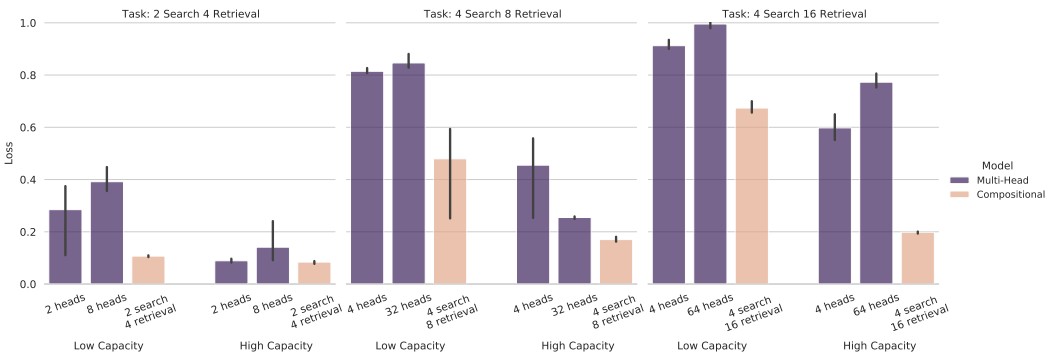

Figure 8: **Performance on Contextual Retrieval Task.** We compare our proposed model against standard Multi-Head attention (lower loss is better) on various setups of the task. Our proposed model outperforms the baseline across various model capacities (low and high) and number of heads.

### B.1 Motivation

While the setting in Figure 1 may look idealistic in the sense that it is very likely that transformer heads do not learn this interpretable single feature functions for search and retrieval, we argue that this rigidity still exists between search and retrieval in a standard multi-head attention framework. To see this, we note that the search component $Search_h$, is parameterized by the query and key matrices $W_{q_h}$ and $W_{k_h}$ respectively and the retrieval component $Retrieval_h$ is parameterized by the value matrices $W_{v_h}$. Both these components lead to computations that are dynamic based on the input but the pairing between search and retrieval is fixed, that is, $Retrieval_h$ takes $Search_h$ as its argument (notice the same $h$ subscript), also highlighted in Equation 5. Thus, whenever there is a need to share retrieval parameterizations across multiple searches, a standard multi-head attention would lead to learning of redundancies because there is no notion of sharing of retrievals between searches.

Contrasting this with the proposed approach, Compositional Attention, we see that now there is a notion of sharing of retrievals for different searches. That is, two different searches can still opt for the same retrieval parameterization, which alleviates the rigidity and redundancy that is explained above. Note that this discussion does not depend on the model's capacity to selectively pick features as is illustrated in Figure 1. This shows that irrespective of what these searches and retrievals learn, the discussed drawbacks of multi-head attention still exist if an optimal solution requires sharing of retrievals across searches. We highlight the motivation through the idealistic example of multiple features solely for ease of explanation and appealing to the fundamental cognitively inspired inductive bias that we try to incorporate.

We emphasize that multi-head attention and the proposed compositional attention are not two separate classes of methods. In fact, our proposed mechanism is a strict superset of multi-head attention and thus presents a more general framework that subsumes the family of multi-head attention. One can see this from Equation 13 where, given enough capacity to represent any $h \times h$ matrix, we recover multi-head attention by setting the number of searches and retrievals as $h$ and having the "Value Scores" matrix as an $h \times h$ identity matrix (or any $h \times h$ permutation matrix in general), with $h$ being the number of heads. Thus our mechanism not only solves the redundancies highlighted in this text but also provides a more general class of attention mechanism.

### B.2 Differences from Existing Work

We propose *Compositional Attention*, a novel attention mechanism aimed at a disentangled computation of search and retrieval. Unlike in multi-head attention, this allows for a flexible and dynamic composition of searches and retrievals.

This is different from MAC and its variants (Hudson & Manning, 2018; Selvakumar et al., 2018) because the proposed algorithm is a completely parallel system without recurrence. Further, we see that in MAC, disentanglement is driven by privileged information; i.e. through the difference between what a question and image is. This privileged information may not be present across a variety of tasks (eg. language modelling, classification, etc.). Our proposed model, however, does not require privileged information and is therefore easily applicable to a lot of different domains. Further, MAC

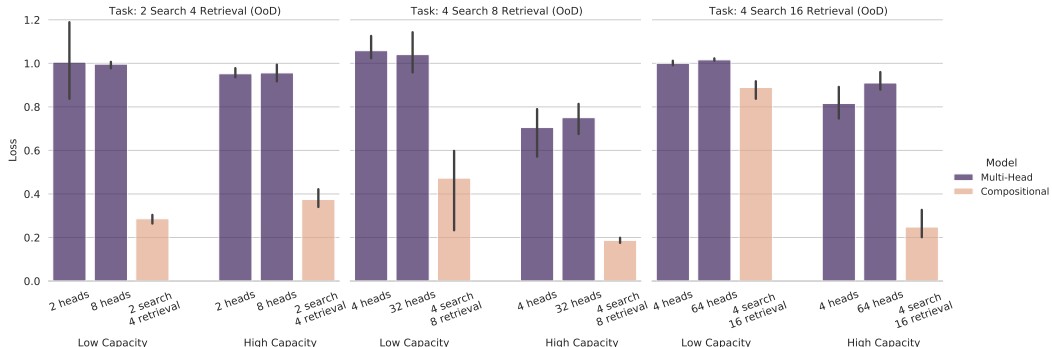

Figure 9: **Performance on OoD Contextual Retrieval Task.** We showcase that our proposed mechanism outperforms standard Multi-Head attention (lower is better) on out of distribution (OoD) variant of the various setups across various model capacities (low and high) and number of heads.

does not have multiple parallel searches and retrievals and thus, our proposed model aims to solve a considerably different problem.

While one may be tempted to think of head pruning (Michel et al., 2019; Voita et al., 2019) as a way of removing redundancies in standard multi-head attention, we stress that the core goal and motivation of our work is considerably different. Pruning of a head essentially means we eliminate a rigid search-retrieval pairing from the learned system as its utility for solving the task is negligible. However, in this work, the redundancy we want to solve is when a sub-part of a head is redundant but not the whole head. That is, when either the search or retrieval part of the head is redundant, but not both. Figure 1 highlights when only a sub-part of the head is redundant and not the whole head, and how compositional attention resolves the problem.

Further, Compositional Attention is different from the various transformer variants Appendix A.2 because it does not aim to solve the quadratic computational bottleneck but instead adds an inductive bias that has not been explored yet. We also note that the proposed model is amenable to the various computation tricks discovered for multi-head attention.

## B.3 ARCHITECTURE DETAILS

The standard transformer model (Vaswani et al., 2017) has a number of layers, where each layer is composed of two components, the multi-head attention (Section 2.1) which is followed by a MLP (Multi-layer perceptron) with a single hidden layer. There are residual connections at the end of the multi-head attention step as well as the MLP.

In this work, we follow Dehghani et al. (2018) and consider the models that have weight sharing across layers. For ease of experiments, we do not consider adaptive stopping criteria. We consider this choice because we want reusable pieces of computations, and Universal Transformers is one step towards that goal.

Our view of transformer models is that different heads perform parallel information retrieval with not only different kinds of searches but also different kinds of retrievals. Information from these parallel retrievals is then jointly processed through a linear layer, followed by another MLP. There are residual connections after the linear layer and the MLP.

For our proposed Compositional variants, we basically replace Multi-Head Attention in the models with Compositional Attention while keeping all the other details the same.

## B.4 MULTIPLE LAYERS AND WEIGHT SHARING

A number of works demonstrate that Transformers with weight sharing are competitive with the standard transformer models (Dehghani et al., 2018; Bai et al., 2019). We also believe that reusing computations provides more pressure on the system to learn meaningful and multi-purpose parameters (eg. it is easier to learn a redundant head if it is used only once vs if it is repeatedly used). One might be tempted to think that increasing the number of layers or removing weight sharing might compensate for the flexibility provided by our proposed system. However, we argue otherwise.

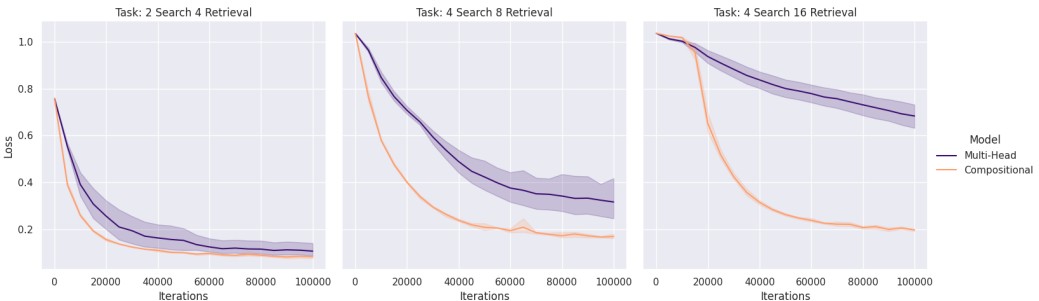

Figure 10: **Convergence on Contextual Retrieval Task.** We see that the proposed mechanism converges faster and works well even in low data regime (low iterations).

Lets assume we have a Transformer model without parameter sharing which has $l$ layers and $h$ heads. Then, the number of unique search-retrieval pairings that can be computed by the model is $lh$ ($h$ if parameter sharing). Contrasting this with compositional attention, we see that the number of unique search-retrieval pairings are actually $lsr$ ($sr$ if parameter sharing) where $s$ is the number of searches and $r$ the number of retrievals. So, if we use a similar number of layers, compositional attention still allows for more combinatorial possibilities to be learnt. Viewed another way, at scale, the proposed mechanism has the potential to reduce the number of layers needed for tasks calling for flexible search and retrievals.

Another important point is that even if we have more layers (with or without parameter sharing), multi-head attention can still only learn a rigid combination between search and retrieval. So, if the task requires dynamic choice from all possible pairings between search and retrieval, the model will have to learn each pairing in separate head combinations, whether it be in the same or future layers. This is because adding more layers does not change the way searches and retrievals are combined, which is what we focus on here.

### B.5 COMPUTATIONAL COMPLEXITY

**Number of Parameters.** We keep the parameter counts within ~5% of each other for the compared models and the same parameter count at 140M parameters for the language modelling experiment. We also stress that our proposed models with fewer retrievals are even more tightly matched and often lower in parameters than the baseline and still outperform them on a number of tasks.

**Training Time.** While Compositional Attention increases the complexity of the model, we note that the training time of proposed models are generally within ~10% of the baseline and hence the added complexity does not impede the model much.

**FLOPs.** We estimate the FLOPs of the proposed model for Equilateral Triangle Detection task using an off the shelf library [2] and see that they are ~10% of each other and the baseline. In particular, we also see that for fewer retrievals, the FLOPs are either the same or lower than the baseline.

**Parallel Computations.** Transformers allow for efficient implementation using GPUs due to parallel computations for each word in the sentence (or each object in the scene). Further, they allow for parallel computation of each head for each word. Correspondingly, in our proposed model, we still do parallel computations for each word in the sentence, and compute the output of different searches in parallel. The only additional complexity is another soft-attention for choice of retrieval for each search. This is also done in parallel for each search and hence we retain all the major efficiencies that Multi-Head attention enjoys on GPUs.

**Amenable to Different Variations.** We note that a lot of the current advances in standard multi-head attention, eg. sparse attention matrix, can be incorporated in the proposed model too. We can also have sparsity on the retrieval end where we can restrict certain searches to pick from a smaller set of retrievals. We believe that these analysis are important future works but out of scope of this paper.

**Complexity vs Combinatorial Advantages.** While we sometimes have more complexity than multi-head attention, this small increase in complexity is often offset by the combinatorial advantage that

---

[2]https://github.com/Lyken17/pytorch-OpCounter

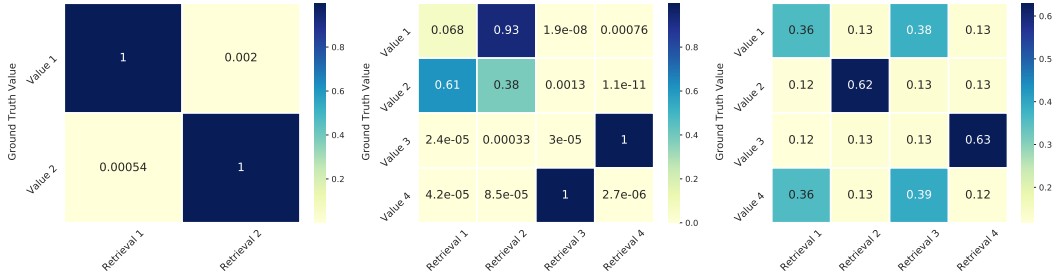

Figure 11: **Specialization plots for the Contextual Retrieval Task.** We plot the attention scores for ground truth retrieval vs learned retrieval for different task setups – **left:** 1 search 2 retrieval, **middle:** 1 search 4 retrieval, and **right:** 2 search 4 retrieval.

we gain. In particular, for $h$ search and retrievals, multi-head attention can only compute $h$ possible search-retrieval pairings while the proposed model can compute $h^2$ possible pairings.

**Controls.** In all our experiments, we control for the number of layers, searches and parameters between the baseline and the proposed model.

### B.6 DETAILS ABOUT SCALING

Suppose our systems have $h$ heads for multihead attention and $s$ searches, $r$ retrievals for compositional attention. Lets assume the input to the system has $N$ tokens. Then, we can see that the number of parameters in multi-head attention is proportional to $3h$ while in compositional attention, it is proportional to $(2s + r)$.

Further, focusing on the highest computational cost of the attention procedure (which are associated with the coefficients of $N^2$ and ignoring the coefficients of terms linear in $N$ ), we see that the coefficient of quadratic complexity is proportional to $2h$ in multi-head attention and $s(1 + r)$ in compositional attention.

This shows that depending on $r$, there can be fewer parameters in the proposed model but the time complexity is strictly higher. This is because we allow for combinatorial more search-retrieval pairings and this cannot be obtained free of cost (no free lunch theorems).

Importantly, if a task that requires $h$ searches ($s = h$) and $h$ retrievals ($r = h$) and a dynamic choice of any search-retrieval pair out of all possibilities (which are $h^2$), then multi-head attention would require $h^2$ heads which leads to $(3h^2)$ parameters and $(2h^2)$ computational cost to fully do this task well while the proposed model would only require $3h$ parameters and $(h^2 + h)$ computational cost, which would be significantly smaller and also more computationally efficient when compared to multi-head attention. We use this exact motivation as best as we could in the Contextual Retrieval Task to showcase the fundamental differences and the parameter efficiency.

### B.7 ABLATIONS

**Retrievals.** For a number of tasks, we keep the number of searches the same as the number of heads in the baseline and then ablate over the number of retrievals. Overall, we notice that most of the models outperform baselines with fewer number of retrievals highlighting the combinatorial advantage of compositional attention. Please refer Table 1, 4, 2, 6, etc. We further ablate on the model capacity and heads while keeping the searches and retrievals fixed for the proposed model. In Table 7, we demonstrate that the head redundancies indeed hurt the performance of the model however the flexible models like compositional attention perform better.

**Compositional-MLP.** As an additional ablation of the retrieval selection mechanism, we aim to replace the dot product attention in Equation 13 for the computation of Value Scores with a Multi-Layer Perceptron (MLP) that takes the retrieval query and key as input and outputs the retrieval attention score. Our MLP is only a linear network but we still generally notice decent performance, as highlighted in Table 8 and 9.

### B.8 LIMITATIONS AND FUTURE WORK

Compositional attention, while more flexible than standard multi-head attention, is still conditional on the mechanism's ability to efficiently perform value retrieval. In particular, we do not explicitly

impose a bottleneck on the different value matrices to pressure the system into learning diverse retrievals, which when combined with joint learning of the retrieval selection mechanism can lead to sub-optimal solutions from gradient-based learning. This is highlighted in Figure 12. Potential future work aims at solving this issue.

Another interesting direction for future development would be to dynamically restrict the set of retrievals accessibly by a search to a sparse number in between 1 (standard Transformer) and all (our compositional attention). This would allow to fine tune the tradeoff between complexity and expressivity for the task at hand.

## C  EXPERIMENTS

In this section, we provide further details about the tasks outlined in Section 4. We also provide task-specific architecture details below.

### C.1  CONTEXTUAL RETRIEVAL TASK

We define the main details for the task in Section 4.1 and provide the additional required details and motivation below.

**Motivation.** Our aim was with this experiment was to design an experiment that consists of multiple objects, each of which have a number of scalar features. Corresponding to each object, we have a set of ground truth searches defined by the *search features* and a set of ground truth retrievals defined by *retrieval features*. The choice for which retrieval to pick is decided by the *retrieval preference* of the object. This is very similar to real world scenarios where objects have a number of features, and search and retrieval can be done about any feature depending on the context. Here, we make our task easier by defining a fixed search for all objects (i.e. no search preference), providing the retrieval preference as a one-hot signal and considering only independent scalar features.

| Train Set | Test Set |
|---|---|
| (0, 0), (0, 1), (0, 2), (0, 3) | (2, 1) |
| (1, 0), (1, 1), (1, 2), (1, 3) | (2, 3) |
| (2, 0), (2, 2) | (3, 1) |
| (3, 0), (3, 2) | (3, 3) |

Table 5: **Contextual Retrieval Task OoD Setup.** For 2-Search-4-Retrieve variant there are $4^2$ unique value combinations that define tasks. For the OoD setup, we pick fraction of tasks for training set and the rest for the test set.

**Dataset.** We consider a number of search-retrieval combination settings for this task. We draw $\alpha_s \sim U(-1, 1)$ and the features $z, \tilde{z} \sim \mathcal{N}(0, 1)$ and consider the setups of 2 search 4 retrievals, 4 search 8 retrievals and 4 search 16 retrievals.

**OoD Setup.** For the OoD setup, we show a certain subset of all possible value combinations for training, and the rest for zero shot testing. We illustrate an example of train and test retrieval combinations in Figure 5 for the 2 search 4 retrieval variant.

**Implementation Details.** For each variant of the task, we ablate on a number of heads and transformer hidden dimensions. We opt to exclude the residual network because it isn't needed for the task, purely from the data generating perspective. For the search, we explicitly mask out the diagonal since the ground truth search is defined as $\arg\min_{j \neq i}$ and along with all the values, we also feed in an extra value from each hidden dimension because it contains information about the retrieval preferences.

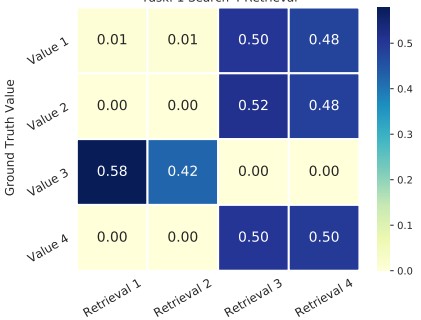

Figure 12: **Contextual Retrieval Task**. Failure case of proposed mechanism in specializing retrievals in 1 search – 4 retrieval setup.

**Quantitative Results.** We highlight the results of our ablations in Table 6 which highlights that not only Compositional Attention outperforms the standard Multihead Attention but also does so across a number of hyperparameter choices. In particular, we often see that the proposed attention mechanism outperforms the baseline even with lower number of parameters. Further, since we generate data on the fly, low training iterations correspond to low data regime on this task. Figure 10

| Ground Truth | Algorithm | Dimension | # Params | Number of Heads | Number of Searches | Number of Retrievals | Loss (in-distribution) | Loss Ood |
|---|---|---|---|---|---|---|---|---|
| 2 Search 4 Retrieval | Multi-head | 64 | 30k | 2 | - | - | 0.28 | 1.00 |
| | | | 28k | 4 | - | - | 0.31 | 0.96 |
| | | | 27k | 8 | - | - | 0.39 | 0.96 |
| | | 128 | 117k | 2 | - | - | 0.08 | 0.95 |
| | | | 109k | 4 | - | - | 0.09 | 0.89 |
| | | | 105k | 8 | - | - | 0.14 | 0.95 |
| | Compositional | 64 | 30k | - | 2 | 4 | 0.10 | 0.28 |
| | | 128 | 119k | - | 2 | 4 | 0.08 | 0.37 |
| 4 Search 8 Retrieval | Multi-head | 128 | 113k | 4 | - | - | 0.81 | 1.05 |
| | | | 109k | 8 | - | - | 0.80 | 0.93 |
| | | | 106k | 32 | - | - | 0.84 | 1.04 |
| | | 256 | 439k | 4 | - | - | 0.54 | 0.89 |
| | | | 422k | 8 | - | - | 0.57 | 0.83 |
| | | | 410k | 32 | - | - | 0.61 | 0.86 |
| | | 512 | 1.7M | 4 | - | - | 0.45 | 0.70 |
| | | | 1.6M | 8 | - | - | 0.24 | 0.67 |
| | | | 1.6M | 32 | - | - | 0.25 | 0.75 |
| | Compositional | 128 | 118k | - | 4 | 8 | 0.48 | 0.47 |
| | | 256 | 458k | - | 4 | 8 | 0.31 | 0.18 |
| | | 512 | 1.8M | - | 4 | 8 | 0.17 | 0.18 |
| 4 Search 16 Retrieval | Multi-head | 128 | 118k | 4 | - | - | 0.91 | 1.00 |
| | | | 112k | 16 | - | - | 0.95 | 1.00 |
| | | | 110k | 64 | - | - | 0.99 | 1.01 |
| | | 256 | 449k | 4 | - | - | 0.77 | 0.91 |
| | | | 424k | 16 | - | - | 0.84 | 0.96 |
| | | | 418k | 64 | - | - | 0.92 | 0.98 |
| | | 512 | 1.7M | 4 | - | - | 0.59 | 0.81 |
| | | | 1.6M | 16 | - | - | 0.67 | 0.83 |
| | | | 1.6M | 64 | - | - | 0.77 | 0.90 |
| | Compositional | 128 | 116k | - | 4 | 16 | 0.67 | 0.88 |
| | | 256 | 442k | - | 4 | 16 | 0.39 | 0.21 |
| | | 512 | 1.7M | - | 4 | 16 | 0.20 | 0.24 |

Table 6: **Performance on the Contextual Retrieval task.** Performance for different number of searches and retrievals in ground truth data. Ablations are done on the number of heads and dimensionality of the transformer dimension.

demonstrates that Compositional Attention not only converges faster but also does better in low-data regime.

**Qualitative Results.** We visualize the activation pattern of our proposed model for the different task setups in Figure 11 which shows that the model often specializes its retrievals according to the ground-truth (up to a permutation). We further see in Figure 12 that sometimes the specialization does not occur which is what we discuss in Section 5.

## C.2 SORT-OF-CLEVR

**Dataset.** Sort-of-CLEVR consist of $10k$ images, where each image is accompanied with 10 *Non-relational* and 10 *Relational* questions. Non-relational questions tests the model ability to focus on the properties of a "single" object like shape, horizontal location and vertical location. A few examples for non-relational questions are (i)*what is the shape of green object?* (ii) *what is the horizontal location of the red object?* (iii) *what is the vertical location of yellow object?* etc. On the other hand, the relational questions tests the model's ability to reason about the attributes of one or more objects. A few examples for relational questions are (i) *What is the shape of the object that is farthest from the green object?*, (ii) *What is the shape of the object that is closest to the red object?*". A sample of the dataset is shown in Figure 13 Each input image is of size $75 \times 75$ and it contains 6 square or circular objects. Each object is colored using one of these 6 different colors (red, blue, green, orange, yellow, gray) to make them visibly distinct. The accompanied questions (relational and non-relational) are encoded in vector of size 11 where the first six entries encode the color information, the next two encode the question type and the last three encode the question sub-type all in one-hot way.

**Implementation Details.** We use a 4-layered transformer with shared parameters and ablate with transformer dimensions 32, 256 and 512 and ffn dimension as 64, 512, 1024 respectively. We consider baseline with 4 and 8 heads and for the proposed model, we use 4 searches and ablate on 1 - 4 retrievals. We use 32 dimensions for the retrieval query and key dimensions. We train the model

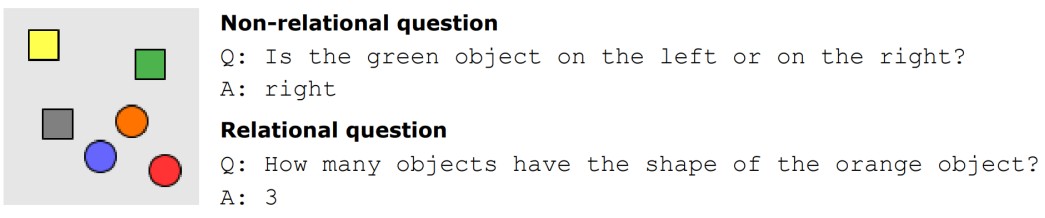

Figure 13: **Sort-of-CLEVR.** Samples from the dataset. Non-relational refers to unary type questions and Relational refers to binary and ternary type questions. Source: Santoro et al. (2017)

| Algorithm | Dimensions | Heads | Unary Accuracy | Binary Accuracy | Ternary Accuracy |
|---|---|---|---|---|---|
| Transformer | 32 | 2 | $66.2_{\pm 8.8}$ | $72.8_{\pm 0.8}$ | $54.5_{\pm 53.6}$ |
| Compositional Transformer | | - | $74.1_{\pm 13.2}$ | $73.7_{\pm 2.0}$ | $53.6_{\pm 0.8}$ |
| Transformer | 256 | 4 | $98.6_{\pm 0.2}$ | $84.4_{\pm 5.3}$ | $64.9_{\pm 3.3}$ |
| | | 8 | $98.5_{\pm 0.2}$ | $84.5_{\pm 6.0}$ | $65.4_{\pm 4.7}$ |
| Compositional Transformer | | - | $98.8_{\pm 0.1}$ | $88.2_{\pm 3.2}$ | $66.9_{\pm 1.8}$ |
| Transformer | 512 | 4 | $98.5_{\pm 0.6}$ | $84.2_{\pm 4.7}$ | $61.5_{\pm 4.8}$ |
| | | 8 | $98.5_{\pm 0.4}$ | $81.5_{\pm 5.0}$ | $62.2_{\pm 4.6}$ |
| Compositional Transformer | | - | $98.9_{\pm 0.3}$ | $84.5_{\pm 5.0}$ | $62.1_{\pm 4.3}$ |

Table 7: **Dimensions and Heads Ablation on Sort of CLEVR.** We perform ablations with increased number of dimensions and heads. For proposed model, we use 2 searches – 2 retrievals for 32 dimensional model and 4 searches – 2 retrievals for other dimensions.

with 0.0001 learning rate for 100 epochs. For all our experiments, we report the mean and standard deviation over 5 seeds.

**Quantitative Results.** Apart from the results presented in the main paper, we also ablate on different capacities and different number of heads in standard transformer. We highlight in Table 7 that our proposed attention mechanism works well across differently sized models consistently and outperforms baselines with more number of heads. We further illustrate in Table 8 that Compositional Attention also outperforms the standard models with fewer searches and performs well even when the retrieval dot-product attention is replaced by the MLP based attention as outlined in the ablations Appendix B.7.

**Qualitative Results.** Here we perform analysis on the learned models on the Sort-of-CLEVR dataset. We average the value scores in Equation 13 over the entire test set and plot value activation statistics of retrievals vs. the type of questions from the task database in Figure 14. We note that there are distinct differences in the activation pattern between unary, binary and ternary type questions.

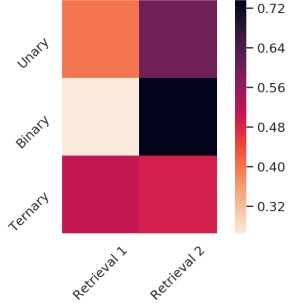

Figure 14: **Sort-of-CLEVR Retrieval Activation.** Activation statistics against the different types of questions in the dataset.

### C.3 EQUILATERAL TRIANGLE DETECTION

This is a binary classification task introduced by Ahmad & Omohundro (2009), where the aim is to decide if the set of three point clusters in the image forms an equilateral triangle. The dataset consists of $64 \times 64$ images with three randomly placed point clusters. For a triangle to be equilateral, the midpoints of these clusters should be equidistant from each other.

| Algorithm | Searches | Retrievals | Unary Accuracy | Binary Accuracy | Ternary Accuracy |
|---|---|---|---|---|---|
| Transformer | 4 | 4 | $98.6_{\pm0.2}$ | $84.4_{\pm5.3}$ | $64.9_{\pm3.3}$ |
|  | 8 | 8 | $98.5_{\pm0.2}$ | $84.5_{\pm6.0}$ | $65.4_{\pm4.7}$ |
| Compositional Transformer | 4 | 1 | $98.7_{\pm0.2}$ | $86.8_{\pm2.8}$ | $66.4_{\pm1.3}$ |
|  |  | 2 | $98.8_{\pm0.1}$ | $88.2_{\pm3.2}$ | $66.9_{\pm1.8}$ |
|  |  | 3 | $98.9_{\pm0.2}$ | $89.8_{\pm1.1}$ | $67.1_{\pm1.5}$ |
|  |  | 4 | $98.6_{\pm0.3}$ | $84.9_{\pm4.5}$ | $64.8_{\pm4.1}$ |
| Compositional Transformer - MLP | 4 | 1 | $98.7_{\pm0.2}$ | $85.8_{\pm4.7}$ | $63.8_{\pm4.0}$ |
|  |  | 2 | $98.8_{\pm0.3}$ | $84.7_{\pm5.8}$ | $65.3_{\pm3.7}$ |
|  |  | 3 | $98.8_{\pm0.15}$ | $84.6_{\pm5.56}$ | $66.0_{\pm2.6}$ |
|  |  | 4 | $98.8_{\pm0.1}$ | $87.8_{\pm2.2}$ | $66.2_{\pm0.7}$ |

Table 8: **Compositional Transformer - MLP Ablation on Sort of CLEVR.** We highlight that our proposed model outperforms the baseline across the different question types even with lower number of searches and/or retrievals even with the MLP ablation where the retrieval attention score is computed by an MLP instead of dot-product acttention.

**Implementation Details.** We follow the same setup as Dosovitskiy et al. (2020) and treat this as a classification task. Each image is split into $4\times4$ patches which is encoded through an MLP and incorporated with position encodings. These are then fed to a 4-layered transformer (with parameter sharing) along with the CLS token which acts as the classification head.

We set the hidden dimension of the transformer as 256 and the ffn dimension as 512. We use 4 heads for the baseline and for the proposed model we use 4 searches and ablate over 1, 2 and 4 retrievals. The retrieval queries and keys are 32 dimensional (used in Equation 13). All the models are trained with a learning rate of 0.0001 for 200 epochs with cosine annealing, similar to Goyal et al. (2021b). For all our experiments, we report the mean and standard deviation over 3 seeds.

**Qualitative Results.** We illustrate in Figure 15 that Compositional Attention not only disentangles search and retrieval but can also allow for tighter pairing between them if need be, depending on the gradient signal. Thus, we see that it leads to increased capacity because the pairing matrix (between search and retrieval) is dynamic, context dependent and can be more than just an identity matrix.

**Quantitative Results.** We see in Table 9 that the proposed compositional attention outperforms the multi-head attention baseline not only across different number of retrievals but also on the MLP (Appendix B.7) ablation where the retrieval selection score is obtained through a simple linear network instead of the dot-product attention mechanism. This shows that even different ways of having this flexible composition of search and retrieval (apart from dot-product based) can lead to improved performance.

| Algorithm | Searches | Retrievals | Test Accuracy |
|---|---|---|---|
| Transformer | 4 | 4 | $93.8_{\pm0.1}$ |
| Compositional Transformer | 4 | 1 | $95.6_{\pm1.2}$ |
|  |  | 2 | $96.7_{\pm0.4}$ |
|  |  | 4 | $97.0_{\pm0.3}$ |
| Compositional Transformer - MLP | 4 | 1 | $96.8_{\pm0.5}$ |
|  |  | 2 | $96.1_{\pm0.6}$ |
|  |  | 4 | $95.5_{\pm1.2}$ |

Table 9: **Performance on Equilateral Triangle Detection.** We perform ablations over the number of retrievals and type of attention mechanism used retrieval. Our proposed method outperforms the baseline over multiple different setups.

Figure 15: **Search-Retrieval Pairing in Equilateral Triangle Detection.** We visualize the activation statistics of learned retrievals (X axis) against learned searches showing that Compositional Attention can also learn tight pairing between search and retrieval.

## C.4 MULTI-TASK IMAGE CLASSIFICATION

**Dataset.** This task is composed of four datasets – CIFAR10, FahsionMNIST, SVHN and Equilateral Triangle Detection. Given an image from one of these datasets, the model is tasked to make the respective classification prediction. We use random crop and horizontal flips as data augmentation

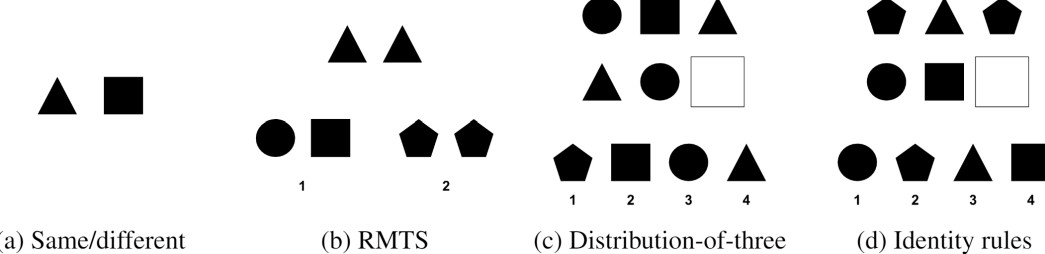

(a) Same/different      (b) RMTS      (c) Distribution-of-three      (d) Identity rules

Figure 16: **Logical Reasoning in ESBN Tasks**. Illustration of the four tasks in the suite. (a) **Same/Different.** Predict whether the two objects are identical or not; they are not in this case, (b) **RMTS.** Match the relation in the context image to the two choices; here option 2 is the right answer since similar to context objects, it also has two identical objects, (c) **Distribution of 3**. Find the missing object with permutations rule; option 2 is the right answer since the square is in the context objects and missing from the second row, and (d) **Identity Rules**. Find the missing object with ABA rule; option 1 is the correct answer since it follows the same rule of identical objects on the edges. Source: Webb et al. (2020)

techniques. This task tests the model's ability to learn multi-purpose representations that can be used for performing well on multiple tasks that potentially share meaningful information.

**Implementation Details.** We train a 2-layered universal transformer with 4 heads. We set the hidden dimension of the transformer as 256 and the ffn dimension as 512. For our proposed model, we keep all the settings as the same, use 4 searches, and ablate over the number of retrievals. The image is cropped into $4 \times 4$ patches and then fed into an encoder and augmented with positional information. For prediction on each dataset, there is a learnable embedding that gets fed into the transformer along with the cropped patches. The retrieval queries and keys are 16 dimensional and we use a learning rate of 0.0001. For our experiments, we report the mean and standard deviation over 3 seeds.

**Quantitative Results.** We showcase in Table 2 that the proposed model outperforms the standard multi-head attention across different number of retrievals.

## C.5 LOGICAL REASONING IN ESBN TASKS

**Dataset.** This is a suite of four tasks – Same/Different, RMTS, Distribution of 3, and Identity Rules (Webb et al., 2020). For each task, the model gets a sequence of objects as input with certain relationship between them. This relationship depends upon the task and based on it, the model is tasked to make a prediction. Further, the dataset consists of an OoD setup where the phrase *m90*, for example, means that training is done on 10 unique objects and testing is done on the remaining 90 objects.

*Same/Different.* The model gets two objects as input and is tasked with predicting whether the two objects are identical or not.

*RMTS.* Similar to Same/Different, we get three pair of objects. The first pair is the context where the first two objects can be either same or different. The second and third pair are the options and the model is tasked with choosing the pair which follows the same relationship as the first pair. That is, if the first pair has identical objects, then the model should pick the pair with identical objects from pair-2 and pair-3.

*Distribution of 3.* We first get three objects as input and then get a permutation of those objects as next inputs with the last object hidden. The model is tasked with predicting which object should fill the last location from a candidate set of four objects that are shown next. In short, the model must choose an object from multiple choices which would make sure that the first three objects and the next three objects are permutations of each other.

*Identity Rules.* The input is given in the form of ABA where A and B are some unique objects. Similar to Distribution of 3, the model then gets two more objects CD and has to choose object C from a candidate set of four choices. That is, the model is tasked to learn the rule of identical peripheral objects in filling up of the final spot.

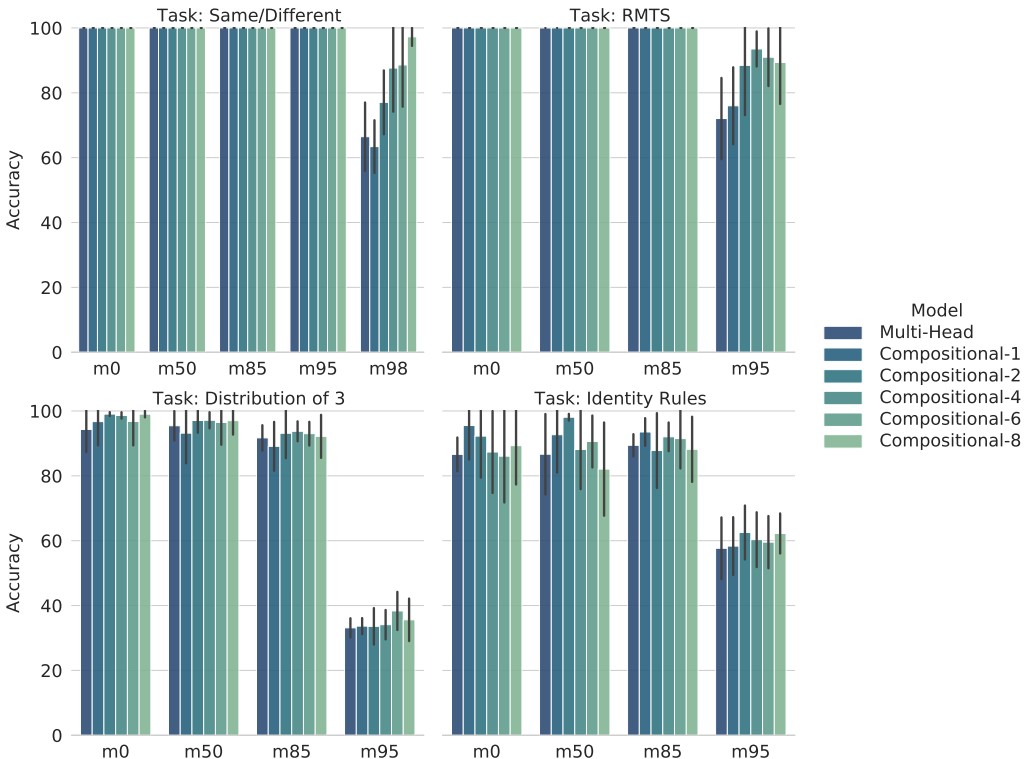

Figure 17: **Logical Reasoning in ESBN Tasks.** We see that compositional attention outperforms multi-head attention baseline over different number of retrievals, especially on Same/Different and RMTS. Compositional-r refers to the proposed model with r retrievals.

We refer the readers to Figure 16 for a visual demonstration of the four tasks.

**Implementation Details.** We follow the setup of Webb et al. (2020) [3] and train a single layered transformer model as baseline and replace multi-head attention with compositional attention in our proposed model. We compare the 8 head baseline model with 8 searches in the proposed model ablated over different retrievals. We use a learning rate of 0.0001 and follow the initialization as well as normalization schemes of the original codebase. We report the mean performance with standard deviation over 10 seeds for each model.

**Quantitative Results.** We notice substantial improvements of the proposed model over the baseline on the Same/Different and RMTS task as illustrated in Figure 17, especially in the OoD setting. We see marginal improvements on the Distribution of 3 and Identity Rules too.

### C.6    SCAN TASK

**Dataset.** The SCAN task (Lake & Baroni, 2018a) is aimed at systematically testing OoD performance of various models. The task requires models to translate natural language inputs, eg. "jump opposite left and walk thrice", into a sequence of actions for a robotic agent to follow, eg. "LTURN LTURN JUMP WALK WALK WALK". In particular, we use the length extrapolation version of this task where models are trained on shorter action sequences and are then evaluated on longer action sequences. Thus, this task tests the capabilities of systematic generalization through compositions of primitive actions. In particular, we follow the task split paradigm as set in Newman et al. (2020); Csordás et al. (2021) which aim to solve certain problems of the original split.

**Implementation Details.** We follow the implementation of Csordás et al. (2021)[4] and train a 3-layered Universal Transformer model with 8 heads. We set the model dimensions as 128, the ffn

---

[3] https://github.com/taylorwwebb/emergent_symbols
[4] https://github.com/robertcsordas/transformer_generalization

dimensions as 256 and the retrieval query and key dimensions as 32. All the models are trained with the learning rate of 0.001.

**Quantitative Results.** The results for the task are showcased in Table 4 which illustrates that the proposed model outperforms the baseline over multiple cutoff lengths, showing that the proposed model is able to better generalize to compositional tasks.

### C.7 LANGUAGE MODELLING

We perform experiments on the WikiText-103 data corpus (Merity et al., 2016) for the language modeling task. The corpus consists of 28,475 articles in its training split and 60 in the validation and test split respectively and the task is to predict probabilities for next words, evaluated by perplexity.

**Quantitative Results:** We use 6-layered transformer models with parameter sharing and perform our experiments on the fairseq codebase (Ott et al., 2019). We plot the *validation perplexity* against epochs in Figure 7 which highlights that our proposed attention mechanism not only outperforms the baseline but also converges faster. Further, we see that our proposed model obtains *test perplexity* $38.8_{\pm 0.0}$ as opposed to baseline's perplexity $39.6_{\pm 0.3}$.

**Implementation Details.** We use the Ott et al. (2019)[5] repository for both pre-processing of the data as well as training. We use a transformer based model with 512 hidden dimensions and 2048 ffn dimensions. Our baseline has 8 heads, and our proposed model has 8 searches and 8 retrievals, and trained with 32 dimensional retrieval queries and keys. We perform a hyperparameter sweep on learning rates – 0.0005, 0.001, 0.002, 0.004 and 0.007. For both the baseline and the proposed method, we choose the model with best validation perplexity (baseline - 0.002 and proposed - 0.004) and then run additional seeds on the chosen learning rate.

---

[5]https://github.com/pytorch/fairseq

