# OpenReview forum: "Compositional Attention: Disentangling Search and Retrieval"
_ICLR.cc/2022/Conference — ICLR 2022 Spotlight_

### Official Review · Reviewer_TCch · 2021-11-02

**Correctness:** 4
**Technical Novelty And Significance:** 3
**Empirical Novelty And Significance:** 3
**Recommendation:** 8
**Confidence:** 4

**Main Review:**

### Strengths

- The paper identifies a current limitation of multi-head attention, and proposes an interesting method to overcome it.

- The paper shows results on a wide variety of experimental tasks.

- The paper is well-written and thorough.

- The illustrative experiment proposed in the paper is an intuitive, clear example of what types of problems Compositional Attention can help with


### Concerns

I would imagine that a Transformer without compositional search/retrieval could possibly compensate for that lack of single-operation compositionality through the stacked computation from the depth of the network.  However, most of the experiments use Universal Transformers as a baseline (or a single layer for the illustrative tasks).  Since the parameters are shared between layers, it seems like that might be a limitation compared to non-shared parameters when the focus is on compositional tasks.  Have you compared to Transformers without shared parameters between layers?

The compositional attention adds overhead compared to a standard MHA.  Although the parameter efficiency argument is well made, there is little analysis of the computational tradeoffs.  How do the Compositional Attention models perform when matched against the baselines for runtime rather than parameter counts? (I understand that is a fraught task given different hardware optimizations/settings between models, but it is still an important consideration)

On a related note, how might Compositional Attention scale to larger models with more searches and queries and more layers?  Does the runtime get much worse than the ~10% reported in the appendix when the models are closer to some of the bigger models reported in the literature?  I’m trying to evaluate how to think of the tradeoff between heavier attention operations run fewer times versus simpler operations with more depth.

**Summary Of The Paper:**

This paper proposes a mechanism for dynamically assigning query-key attention masks (searches) to one of several value matrices (retrievals), avoiding the rigid assignment of search to retrieval inherent in standard multi-head attention.

**Summary Of The Review:**

The paper presents an interesting idea, based on the promising property of compositionality, to increase the power of a single MHA computation and a resulting transformer architecture.  However, I am not quite convinced by the baselines and the analysis of the computational complexity.  I am happy to raise my score if my above concerns are addressed.


### Update after author comments

Thank you for the clear, detailed response and the additional experiments.  You have addressed my concerns, and I will raise my score.  Please consider including the results of the additional experiments and the related discussion of compositionality vs depth in the paper itself, even if just in an appendix.

---

> ### Author Response · Authors · 2021-11-17
> **Author Response Part 1/2**
>
> We would like to thank the reviewer for the detailed feedback. Below, we will elaborate on some of the comments pointed out, and will describe new additions to our submission based on these comments. We would be more than happy to provide any missing details.
>
> **Regarding Multiple Layers**
>
> We thank the reviewer for this comment. Stacked transformer layers with shared parameters are a great way to reuse computations. We are in agreement with the sentiment of accurately accounting for multiple layers with and without parameter sharing, which we illustrate in detail here.
>
> Most of our experiments actually use multiple layers but with parameter sharing for both the proposed and the baseline models since it puts more pressure on learning reusable multi-purpose parameterizations For the illustrative Contextual Retrieval Task, we stick to a single layer because the task definition only requires one layer of computation. Through our original experiments, we can see that with parameter sharing, compositional attention (CA) outperforms multi-head attention (MHA) even with multiple layers.
>
> Before turning to new empirical comparisons, however, we wish to highlight important scaling schemes while comparing multi-head attention and compositional attention with and without parameter sharing.
>
> Let's assume we have a Transformer model without parameter sharing which has $l$ layers and $h$ heads. Then, the number of unique search-retrieval pairings that can be computed by the model is $(l\times h)$ ($h$ if parameter sharing). Contrasting this with CA, we see that the number of unique search-retrieval pairings are actually $(l\times s\times r)$ ($(s\times r)$ if parameter sharing) where $s$ is the number of searches and $r$ the number of retrievals. So, if we use a similar number of layers, compositional attention still allows for more combinatorial possibilities to be learnt. Viewed another way, at scale, CA has the potential to reduce the number of layers needed for tasks calling for flexible search and retrievals. However we leave a full scaling investigation for future work.
>
> Another important point is that even if we have more layers (with or without parameter sharing), MHA can still only learn a rigid combination between search and retrieval. So, if the task requires dynamic choice from all possible pairings between search and retrieval, the model will have to learn each pairing in  separate head combinations, whether it be in the same layer or in further layers. This is because adding more layers does not change the way searches and retrievals are combined, which is what we focus on here.
>
> **Comparison to Transformers without Weight Sharing**
>
> A number of works demonstrate that Transformers with weight sharing are competitive with the standard transformer models (Dehghani et. al; 2018, Bai et. al; 2019). We also believe that reusing computations provides more pressure on the system to learn meaningful and multi-purpose parameters (eg. it is easier to learn a redundant head if it is used only once vs if it is repeatedly used).
>
> However, following the reviewer’s recommendation, we perform experiments on models without parameter sharing and still see that the proposed model outperforms multi-head attention. The performance of the models are showcased below:
>
> *Contextual Retrieval Task*
>
> We train a 2-layered multi-head and compositional model without weight sharing and still see that the proposed mechanism provides improvements over the baseline.
>
> Format - *Model: Regression Loss*
>
> *Transformer:* 1.01
>
> *Compositional:* **0.59**
>
> *(Lower is better)*
>
> For plot, please refer to: https://i.ibb.co/k8Jh5Jw/CRT-norm.png
>
> *Sort-of-CLEVR*
>
> We train a 4-layered multi-head and compositional model without weight sharing. We see that the proposed mechanism still outperforms the baseline even without weight sharing which shows that the benefits are not just because of weight sharing.
>
> Format - *Model: Unary Accuracy, Binary Accuracy, Ternary Accuracy*
>
> *Transformer:* 94.9%, 74.8%, 55.8%
>
> *Compositional-1:* **98.2%**, **82.5%**, **62.4%**
>
> *Compositional-2:* **98.2%**, 79.3%, 58.3%
>
> *Compositional-4:* 98.1%, 77.1%, 57.3%
>
> *(Higher is better)*
>
> Where *Compositional-r* refers to the proposed mechanism with *r* number of retrievals. The plots for the performances can be seen below
>
> Unary Accuracy vs Epochs: https://i.ibb.co/kJqZVQF/unary.png
>
> Binary Accuracy vs Epochs: https://i.ibb.co/ZHs1ghK/binary.png
>
> Ternary Accuracy vs Epochs: https://i.ibb.co/z4g5bkj/ternary.png
>
> *(Higher is better)*
>
> From these ablations, we can see that even by removing parameter sharing, compositional attention still provides improvements over multi-head attention, and thus having more layers or removing parameter sharing both do not solve the problem.
>
> *continued in the next comment.*

---

> > ### Author Response · Authors · 2021-11-17
> > **Author Response Part 2/2**
> >
> > **Matching models on runtime instead of Parameter Counts**
> >
> > We thank the reviewer for their insistence on this point, as we believe it considerably strengthens our result. As per the reviewer’s request, we provide plots of unary, binary, and ternary accuracies of the various models without parameter sharing against their relative wall time. We run all these models on the same cluster and the same GPU type (RTX8000) but there may be other differences between the specific machines that we do not take into account.
> >
> > *Sort-of-CLEVR*
> >
> > Unary Accuracy vs Wall Time: https://i.ibb.co/0sxcQ9z/unary-time.png
> >
> > Binary Accuracy vs Wall Time: https://i.ibb.co/df2MrFP/binary-time.png
> >
> > Ternary Accuracy vs Wall Time: https://i.ibb.co/QNRwCJG/ternary-time.png
> >
> > *(Higher is better)*
> >
> > *Contextual Retrieval Task*
> >
> > We also provide plots of Contextual Retrieval Task with regression loss plotted against iterations. While not exactly wall time, we can see by the large margin between performances that even on the worst case 10% wall time delay, the proposed model would still outperform the baseline.
> >
> > Performance vs Iterations: https://i.ibb.co/nrKpD18/CRT.png
> >
> > *(Lower is better)*
> >
> > We see that even on wall time, CA outperforms MHA.
> >
> > **Scaling Compositional Transformers**
> >
> > We thank the reviewer for their question about scaling complexity. The 10% statistic that we provide is the worst case over all our experiments, including our biggest model with 140M parameters and 6 layers. We provide analysis on the computational complexity and scaling properties below.
> >
> > Suppose our systems have $h$ heads for MHA and $s$ searches, $r$ retrievals for CA. Further, we assume the input to the system has $N$ tokens. Then, focusing on the highest computational cost of the attention procedure (which are associated with the coefficients of $N^2$ and ignoring the coefficients of terms linear in $N$), we can see that
> >
> > *Multi-Head Attention*
> >
> > Number of Parameters is proportional to $3h$
> >
> > Coefficients of quadratic complexity is proportional to $2h$
> >
> > *Compositional Attention*
> >
> > Number of Parameters is proportional to $(2s + r)$
> >
> > Coefficients of quadratic complexity is proportional to $s(1 + r)$
> >
> > The above shows that depending on $r$, there can be fewer parameters in the proposed model but the time complexity is strictly higher. This is because we allow for combinatorial more search-retrieval pairings and this cannot be obtained free of cost (no free lunch theorems).
> >
> > Importantly, if a task that requires $h$ searches ($s = h$) and $h$ retrievals ($r = h$) and a dynamic choice of any search-retrieval pair out of all possibilities (which are $h^2$), then multi-head attention would require $h^2$ heads which leads to $(3h^2)$ parameters and $(2h^2)$ computational cost to fully do this task well while the proposed model would only require $(3h)$ parameters and $(h^2 + h)$ computational cost, which would be significantly smaller and also more computationally efficient when compared to multi-head attention. We use this exact motivation as best as we could in the Contextual Retrieval Task to showcase the fundamental differences and the parameter efficiency.
> >
> > Finally, we would like to remind the reviewer that our analysis on multiple layered transformer models without parameter sharing essentially showcase that the benefits of the proposed method are more far reaching than just increasing layers in the baseline and thus running the simpler operations with more depth won’t provide the same benefits as those obtained from compositional attention, which are a bit more computationally intensive but provide the extra flexibility required if the tasks under consideration have such a compositional search-retrieval pairing requirement.
> >
> > In summary,  we emphasize that multi-head attention and the proposed compositional attention are not two separate classes of methods. In fact, our proposed mechanism is a strict superset of multi-head attention and thus presents a more general framework that subsumes the family of multi-head attention. One can see this from *Equation (13)* where, given enough capacity to represent any $h\times h$ matrix, we recover multi-head attention by setting the number of searches and retrievals as $h$ and having the *“Value Scores”* matrix as an $h \times h$ identity matrix (or any $h\times h$ permutation matrix in general), with $h$ being the number of heads. An interesting direction for future development would be to restrict the set of retrievals a search can pair to to some sparse number in between 1 (standard multi-head attention) and all (our compositional attention). This would allow to fine tune the tradeoff between complexity and expressivity for the task at hand. We add a discussion point about this future work in *Appendix B.6* (Page 18).
> >
> > We hope that we have resolved all of the reviewer’s concerns regarding our work. We would be more than happy to provide further details and resolve any other concerns that the reviewer may have.

---

> > > ### Author Response · Authors · 2021-11-21
> > > **Thank you for the update**
> > >
> > > We would like to thank the reviewer for taking the time to go through the rebuttal and for updating their score. We also acknowledge that the discussion with the reviewer led to an overall improvement of the paper and we include different parts of these discussions in the appendix in a revised version of the draft (parts in blue in the pdf).

---

### Official Review · Reviewer_RbM7 · 2021-11-05

**Correctness:** 3
**Technical Novelty And Significance:** 3
**Empirical Novelty And Significance:** 3
**Recommendation:** 6
**Confidence:** 3

**Main Review:**

Strengths:
-  The problem studied in this paper is interesting and well-motivated.
-  The paper points out the shortcomings of rigid search-and-retrieval coupling in standard multi-head attention.
-  The proposed compositional attention mechanism is rational.

Weaknesses:
-  The evaluation tasks selected in this paper are somewhat weird. The author should conduct experiments on some traditional or mainstream tasks  (or settings) of language or vision data.
- The paper should also compare with other variants of multi-head attention mechanisms.
- The paper should explore the effect of the proposed compositional attention mechanism on the latest pre-trained language models.







**Summary Of The Paper:**

The paper studies the multi-head attention mechanism in Transformer. The paper first analysis the potential drawback of the rigid mapping between search and retrieval in standard attention heads, and then proposes compositional attention that disentangles search and retrieval and composes them in a dynamic and context-dependent manner. To evaluate the proposed method, the author conduct experiments on the standard Transformer model and Vision Transformer.

Overall, the paper is clearly written and the problem studied in this paper is interesting and well-motivated. The proposed compositional attention mechanism is rational. The experiment conducted in this paper is somewhat thorough.





**Summary Of The Review:**

The paper provides a somewhat novel insight into the shortcomings of standard multi-head attention, and also proposed a compositional attention mechanism. However, there are still some concerns about the evaluation.

---

> ### Author Response · Authors · 2021-11-17
> **Author Response**
>
> We appreciate the reviewer’s positive and detailed feedback. We attempt to address all the concerns raised by the reviewer and would be more than happy to address any additional questions that the reviewer may have.
>
> **Regarding Tasks**
>
> We would like to point the reviewer to the WikiText-103 language modeling task (Section 4.7), the SCAN dataset task (Section 4.6), ESBN tasks (Section 4.5) and the Sort-of-CLEVR task (Section 4.2) that are some of the mainstream, well established (accepted in various peer-reviewed conferences) tasks in which we use the various existing codebases to build our model on. For the classification-based tasks, we follow the standard setting of Vision Transformer that is built on extracting patches from the images and then running a transformer/proposed model over these extracted patch representations.
>
> **Comparison to other Multi-head attention variants**
>
> Most of the variants of Multi-head attention are focused around reducing the quadratic computational overhead of the search mechanism while here we focus more on an unexplored inductive bias that is not linked with the quadratic cost. We compare with only Multi-head attention to study the benefits of the proposed system without any other confounding factors like sparsity (Child et. al; 2019), randomness (Choromanski et. al; 2020), etc. Further, these additional factors can also be added to the proposed model so it made sense to compare the vanilla version of our system with the vanilla multi-head attention. Since these additions can be made to both the baseline as well as the proposed model, we leave investigation into variants of both for future work.
>
> That being said, we would be happy to compare with other attention variants that are not aimed at speeding up multi-head attention but instead change the attention mechanism fundamentally. To the best of our knowledge, we are not aware of such a mechanism and would be happy if the reviewer could provide us with some pointers.
>
> **Latest pre-trained language models**
>
> The reason we don’t compare with the standard pre-trained language models is that training such models (like GPT-3) requires access to complex infrastructure, multiple optimized GPUs, and other important engineering feats. Since our proposed model needs to be trained from scratch, training a comparable sized Compositional Attention model on comparable dataset size is not something that we possess the bandwidth to pursue. We do believe that a comparison between scaled pre-trained multi-head attention models with scaled compositional attention models would be an important next step but it is currently out of the scope of this paper.
>
> In summary,  we emphasize that multi-head attention and the proposed compositional attention are not two separate classes of methods. In fact, our proposed mechanism is a strict superset of multi-head attention and thus presents a more general framework that subsumes the family of multi-head attention. One can see this from *Equation (13)* where, given enough capacity to represent any $h \times h$ matrix, we recover multi-head attention by setting the number of searches and retrievals as $h$ and having the *“Value Scores”* matrix as an $h \times h$ identity matrix (or any $h\times h$ permutation matrix in general), with $h$ being the number of heads. An interesting direction for future development would be to restrict the set of retrievals a search can pair to to some sparse number in between 1 (standard multi-head) and all (our compositional attention). This would allow to fine tune the tradeoff between complexity and expressivity for the task at hand. We add a discussion point about this future work in *Appendix B.6* (Page 18).
>
> We hope that we have resolved all of the reviewer’s concerns regarding our work. We would be more than happy to provide further details and resolve any other concerns that the reviewer may have.

---

> > ### Author Response · Authors · 2021-11-21
> > **Anything else you would like us to respond to?**
> >
> > Dear reviewer,
> >
> > Since the discussion phase is closing soon, we would like to know if there are any other concerns that we haven't addressed. We will be happy to address them.
> >
> > Thank you.

---

> > ### Comment · Reviewer_RbM7 · 2021-12-02
> > **Response to Rebuttal**
> >
> > I appreciate that the authors answered my questions. I stick to my rating since it reflects my best judgment of the paper.

---

### Official Review · Reviewer_f5jQ · 2021-11-07

**Correctness:** 3
**Technical Novelty And Significance:** 3
**Empirical Novelty And Significance:** 3
**Recommendation:** 6
**Confidence:** 3

**Main Review:**

Strength:

The author presents a comprehensive analysis of the limit of multi-head attention. They point out that the rigid mapping between query-key and value pairs can lead to the learning of redundant query-key matrices and value matrices. Instead, they propose to consider all pairs between query-key and value. They define query-key as a search and value as a retrieval and then introduce new parameters to compute the attention score to compute the weighted sum over the pairs of search to all retrievals.

It compares compositional attention with multi-head attention on a set of tasks including a proof-of-concept contextual retrieval task, Sort-of-CLEVER, logical reasoning tasks, language modeling, etc. Compositional attention has shown consistent improvement on these tasks.

Weakness:

They argue that the complexity overhead of compositional attention is lightweight. More importantly, the overhead can be mitigated by reducing the number of retrievals. The appendix shows the overhead analysis on the contextual retrieval task. It would be much more helpful and convincing to show such analysis on other real world tasks.

The paper lacks the comparison with other related works focusing on head pruning. As the author pointed out, in figure 11, the compositional attention may tend to learn to focus only on several pairs of search and retrieval pairing. This is very relevant to other head pruning works. It’s worth noting what’s the difference between compositional attention and other head pruning techniques.

Another question remaining unclear is what kinds of tasks can benefit from compositional attention? Multi-head attention has been widely adopted in a lot of scenarios, such as pre-training, fine-tuning, various tasks with sufficient data or very little data. Though the author has tried to experiment on a set of tasks, it remains unclear what's the rule-of-thumb to replace compositional attention to multi-head attention.


**Summary Of The Paper:**

This paper proposes a Compositional Attention to replace the standard multi-head attention. It argues that traditional multi-head attention has a rigid mapping between each query-key pair and the associated value. This leads to redundant parameters and less generalizability.


**Summary Of The Review:**

The paper focuses on improving parameter redundancy and generalizability in multi-head attention. The compositional attention is well motivated to mitigate these issues. The paper is well written and has shown improvement of compositional attention. That said, it's lacking analysis of the constraints on computation overhead of compositional attention on more real-world tasks under both sufficient data and low data scenario. It's a bit unclear of the rule-of-thumb to replace the multi-head attention with compositional attention.

---

> ### Author Response · Authors · 2021-11-17
> **Author Response**
>
> We appreciate the reviewer’s positive and detailed feedback. We attempt to address all the concerns raised by the reviewer and would be more than happy to address any additional questions that the reviewer may have.
>
> **Complexity Overhead**
>
> We point the reviewer to *Appendix Section B.4* that illustrates the computational overhead of the proposed model by various complexity metrics. We stress that the numbers provided in the appendix are taken as the worst-case over all our experiments. We also perform ablations on model complexity on Sort-of-CLEVR (*Tables 7-8*) and showcase ablations on the number of retrievals on most of the tasks we consider.
>
> For Language Modelling, both the baseline and the proposed model have 140 million parameters for quantitative metrics. For Sort-of-CLEVR, Multi-Head attention has 684k parameters while Compositional Attention has 670k, 686k, 703k, and 719k parameters for 1, 2, 3, and 4 retrievals respectively. This puts the total number of parameters within ~5% of the baseline. We note that a Multi-Head Attention with ~2M parameters doesn’t outperform the smaller Compositional Attention Models too as they obtain 98.5/84.2/61.5 for unary/binary/ternary splits (*Table 7*). For FLOPs, we note that the baseline has 540M FLOPs while the proposed model has 531M, 556M, 582M, 607M FLOPs for 1-4 retrievals respectively for Equilateral Triangle Detection (computed using https://github.com/Lyken17/pytorch-OpCounter).
>
> **Comparison to Head Pruning**
>
> We thank the reviewer for pointing this out to us as we believe it results in considerable improvements to the paper. We have added a revision to discuss the connections with head pruning (Michel et. al; 2019, Voita et. al; 2019) in *Appendix B.2* (Page 16, last paragraph on the page). The aim of head pruning is to remove heads that are completely redundant and not essential to the task at all. However, here we concentrate on cases where a head as a whole is not redundant but one of its components (either search or retrieval but not both) that is learned is redundant. Note that in *Figure 1*, all of the proposed learned heads are useful but there is still redundancy in the sub-parts of the heads.
>
> **Kind of Tasks for Compositional Attention**
>
> The main benefit of the proposed model is the dynamic selection of retrievals for each search. Hence, the biggest advantage of this model is in cases where it is required to dynamically retrieve different information based on a common search (as highlighted in *Figure 1*). For example, if the task just required one unique retrieval per search (eg. if we search an object by color and always query its location) then there is no advantage of having the flexibility of dynamic retrieval selection.
>
> While we don’t explicitly test for the performance of our model in low-data settings, we can see in *Figure 7* that the proposed model converges faster. We provide additional plots of performance against epochs/training iterations as a proxy to highlight the speed of convergence and performance of the model in the low data regime for the Contextual Retrieval Task ( since the task generates new data per iteration, low number of iterations refer to fewer unique data-points seen ). This can be seen in the following plot: https://i.ibb.co/nrKpD18/CRT.png
>
> For most real-world applications and object-centric tasks, we often require a dynamic selection of information to retrieve per search. It would be very interesting to scale this approach to see how it performs when compared to the pre-trained Multi-head models but this requires a lot of compute capacity and other engineering challenges that are out of the scope of this work.
>
> In summary,  we emphasize that multi-head attention and the proposed compositional attention are not two separate classes of methods. In fact, our proposed mechanism is a strict superset of multi-head attention and thus presents a more general framework that subsumes the family of multi-head attention. One can see this from *Equation (13)* where, given enough capacity to represent any $h\times h$ matrix, we recover multi-head attention by setting the number of searches and retrievals as $h$ and having the *“Value Scores”* matrix as an $h \times h$ identity matrix (or any $h\times h$ permutation matrix in general), with $h$ being the number of heads. An interesting direction for future development would be to restrict the set of retrievals a search can pair to to some sparse number in between 1 (standard multi-head attention) and all (our compositional attention). This would allow to fine tune the tradeoff between complexity and expressivity for the task at hand. We add a discussion point about this future work in *Appendix B.6* (Page 18).
>
> We hope that we have resolved all of the reviewer’s concerns regarding our work. We would be more than happy to provide further details and resolve any other concerns that the reviewer may have.

---

> > ### Author Response · Authors · 2021-11-21
> > **Anything else you would like us to respond to?**
> >
> > Dear reviewer,
> >
> > Since the discussion phase is closing soon, we would like to know if there are any other concerns that we haven't addressed. We will be happy to address them.
> >
> > Thank you.

---

### Official Review · Reviewer_kYB6 · 2021-11-10

**Correctness:** 4
**Technical Novelty And Significance:** 3
**Empirical Novelty And Significance:** 3
**Recommendation:** 8
**Confidence:** 3

**Main Review:**

Intuitively, the suggested method makes sense: it provides additional flexibility in the attention mechanism, since now the same retrievers can be re-used by different searchers, and since a search is not confined to a single retriever. Such added flexibility and re-usability could improve the generalization abilities of the attention mechanism.

The authors do a good job of evaluating the proposed mechanism, with a diverse selection of tasks: Reasoning over images (sort of clevr), logical reasoning, SCAN (length split) and language modeling and an additional specifically-designed task that requires contextual retrieval. For some of these datasets, models are evaluated on OOD samples. Since generalization seems to be the main motivation for the proposed architecture design, adding additional OOD tests where possible (e.g. as far as I understand CLEVR was only tested in i.i.d setting) could have made the results stronger. Nevertheless, experiments convincingly show an improvement in almost all cases.

My main concern regarding the experiments is that in all cases, parameters were shared between all transformer layers. Since in most common usages of the models parameters are not shared, it seems important to test how well does this method work with non-shared parameters - would the compositional attention still work well with different search/retrieve parameters?


**Summary Of The Paper:**

The proposed mechanism disentangles the search and retrieval from the transformers architecture, where the retriever is no longer tied to a specific retriever that is in the same head - instead, there are multiple searches available, and multiple retrievers. Conditioned on the input, each search is "tied" to a (soft) single retriever. Then all results from all searches are concatenated.

**Summary Of The Review:**

This paper proposes a modification to the transformer's attention mechanism, which shows improvement on multiple evaluation benchmarks of various types. I find this proposal to be convincingly useful, although evaluation on more o.o.d splits and experiments with non-shared parameters could have made results stronger.

---

> ### Author Response · Authors · 2021-11-17
> **Author Response**
>
> We appreciate the reviewer’s positive and detailed feedback. We attempt to address all the concerns raised by the reviewer and would be more than happy to address any additional questions that the reviewer may have.
>
> **Main Motivation**
>
> We would like to highlight that the main motivation of our work goes beyond OoD generalization. We propose a more flexible and larger class of attention based systems which we call compositional attention (CA). This mechanism actually subsumes multi-head attention as a specific case. We then test on various tasks whether this added flexibility of compositional search-retrieval is actually beneficial in both in-distribution as well as out of distribution settings. Through these experiments, we conclude that indeed having this flexibility helps in a number of cases, especially in object-based reasoning systems (as was hypothesized in *Figure 1*).
>
> Thus, we believe our main motivation is not just OoD generalization but the proposal of a more general class of mechanisms that often lead to better OoD (and in-distribution) generalization as a consequence of this added flexibility.
>
> **Comparison to Transformers without Weight Sharing**
>
> A number of works demonstrate that Transformers with weight sharing are competitive with the standard transformer models (Dehghani et. al; 2018, Bai et. al; 2019). We also believe that reusing computations provides more pressure on the system to learn meaningful and multi-purpose parameters (eg. it is easier to learn a redundant head if it is used only once vs if it is repeatedly used).
>
> However, following the reviewer’s recommendation, we perform experiments on models without parameter sharing and still see that the proposed model outperforms multi-head attention. The performance of the models are showcased below:
>
> *Sort-of-CLEVR*
>
> We train a 4-layered multi-head and compositional model without weight sharing. We see that the proposed mechanism still outperforms the baseline even without weight sharing which shows that the benefits are not just because of weight sharing.
>
> Format - *Model: Unary Accuracy, Binary Accuracy, Ternary Accuracy*
>
> *Transformer:* 94.9%, 74.8%, 55.8%
>
> *Compositional-1:* **98.2%**, **82.5%**, **62.4%**
>
> *Compositional-2:* **98.2%**, 79.3%, 58.3%
>
> *Compositional-4:* 98.1%, 77.1%, 57.3%
>
> *(Higher is better)*
>
> Where *Compositional-r* refers to the proposed mechanism with *r* number of retrievals. The plots for the performances can be seen below
>
> Unary Accuracy vs Epochs: https://i.ibb.co/kJqZVQF/unary.png
>
> Binary Accuracy vs Epochs: https://i.ibb.co/ZHs1ghK/binary.png
>
> Ternary Accuracy vs Epochs: https://i.ibb.co/z4g5bkj/ternary.png
>
> *(Higher is better)*
>
> *Contextual Retrieval Task*
>
> We train a 2-layered multi-head and compositional model without weight sharing and still see that the proposed mechanism provides improvements even without weight sharing.
>
> Format - *Model: Regression Loss*
>
> *Transformer:* 1.01
>
> *Compositional:* **0.59**
>
> *(Lower is better)*
>
> For plot, please refer to: https://i.ibb.co/k8Jh5Jw/CRT-norm.png
>
> *(Lower is better)*
>
> From these ablations, we can see that even by removing parameter sharing, compositional attention still provides improvements over multi-head attention.
>
> In summary,  we emphasize that multi-head attention and the proposed compositional attention are not two separate classes of methods. In fact, our proposed mechanism is a strict superset of multi-head attention and thus presents a more general framework that subsumes the family of multi-head attention. One can see this from *Equation (13)* where, given enough capacity to represent any $h\times h$ matrix, we recover multi-head attention by setting the number of searches and retrievals as $h$ and having the *“Value Scores”* matrix as an $h \times h$ identity matrix (or any $h \times h$ permutation matrix in general), with $h$ being the number of heads. An interesting direction for future development would be to restrict the set of retrievals a search can pair to to some sparse number in between 1 (standard multi-head) and all (our compositional attention). This would allow to fine tune the tradeoff between complexity and expressivity for the task at hand. We add a discussion point about this future work in *Appendix B.6* (Page 18).
>
> We hope that we have resolved all of the reviewer’s concerns regarding our work. We would be more than happy to provide further details and resolve any other concerns that the reviewer may have.

---

> > ### Author Response · Authors · 2021-11-21
> > **Anything else you would like us to respond to?**
> >
> > Dear reviewer,
> >
> > Since the discussion phase is closing soon, we would like to know if there are any other concerns that we haven't addressed. We will be happy to address them.
> >
> > Thank you.

---

> > > ### Comment · Reviewer_kYB6 · 2021-11-26
> > > **Thank you -**
> > >
> > > my concerns have indeed been addressed.

---

### Comment · Area_Chair_hzax · 2021-11-26
**Discussion**

Dear reviewers,
Overall reviews seem quite positive - could you make sure you read the author response and other reviews to see if you would like to amend your review in any way?

---

### Decision · Program_Chairs · 2022-01-20

**Decision:**

Accept (Spotlight)

**Comment:**

This paper identifies a limitation with current attention in transformers where they scoring with query-key pairs is strongly tied to retrieving the value and proposes a more flexible configuration that subsumes the previous setup but provides more flexibility. The authors shows this leads to improvements in various settings.

Overall, all reviewers seem to agree there is interesting insight and results in this paper and it merits publication. Also the discussion helped stress important points regarding weight sharing and more. One concern is that the model was not evaluated on standard NLP/vision datasets (I assume alluding to GLUE/SuperGlue/SQuAD, etc.), and authors seem to hint that pre-training this is an issue for them computationally. This leaves open whether this indeed can and should replace the standard attention mechanism across the board, but is still very worthy of publication.